# Fair Representation Learning with Controllable High Confidence Guarantees via Adversarial Inference

**Yuhong Luo**
Rutgers University
New Brunswick, NJ, USA
y.luo@rutgers.edu

**Austin Hoag**
Sony AI
New York, NY, USA
austin.hoag@sony.com

**Xintong Wang**
Rutgers University
New Brunswick, NJ, USA
xintong.wang@rutgers.edu

**Philip S. Thomas**
University of Massachusetts
Amherst, MA, USA
pthomas@cs.umass.edu

**Przemysław A. Grabowicz [1,2]**
[1]University College Dublin, Ireland
[2]University of Massachusetts, Amherst, MA, USA
przemek.grabowicz@ucd.ie

## Abstract

Representation learning is increasingly applied to generate representations that generalize well across multiple downstream tasks. Ensuring fairness guarantees in representation learning is crucial to prevent unfairness toward specific demographic groups in downstream tasks. In this work, we formally introduce the task of learning representations that achieve high-confidence fairness. We aim to guarantee that demographic disparity in every downstream prediction remains bounded by a *user-defined* error threshold $\varepsilon$, with *controllable* high probability. To this end, we propose the ***F**air **R**epresentation learning with high-confidence **G**uarantees* (FRG) framework, which provides these high-confidence fairness guarantees by leveraging an optimized adversarial model. We empirically evaluate FRG on three real-world datasets, comparing its performance to six state-of-the-art fair representation learning methods. Our results demonstrate that FRG consistently bounds unfairness across a range of downstream models and tasks. The source code for FRG is available at: https://github.com/JamesLuoyh/FRG.

## 1 Introduction

In every prediction task, machine learning algorithms assume two distinct roles: the data producer and the data consumer [19, 48, 75]. The data consumer's role is to make accurate predictions using the data provided by the data producer. While the data producer may distribute raw data, it is common to generate new representations via *representation learning* for the input data that are used as predictors in downstream tasks. When multiple data consumers' prediction tasks involve inputs of the same type, such as natural language text or images, the data producer can generate *general representations* that are predictive to multiple subsequent tasks. This is an increasing trend with examples including the Variational Auto-Encoder (VAE) [38] or recent language models such as BERT [16] and GPT-4 [58], which are widely used as bases for downstream text classification tasks [13].

While representation learning can benefit various downstream predictions, it is also susceptible to the risk of producing unintended or undesirable behaviors in the downstream tasks, specifically, generating predictions that are unfair toward disadvantaged demographic groups. Especially in critical

39th Conference on Neural Information Processing Systems (NeurIPS 2025).

domains, such as loan underwriting [9], hiring [53] and criminal sentencing [4], the consequences of algorithmic bias may severely impact individuals. To address these concerns, researchers have proposed fair representation learning (FRL), emphasizing that fairness should be the responsibility of the data producer who generates the representations [48, 75], rather than the data consumer who uses them. By ensuring fairness at the representation level, the data producer guarantees fairness across all downstream tasks, allowing the representations to be safely used by any data consumer.

Extensive prior work in FRL has shown effectiveness in promoting fairness for specific downstream tasks. Some methods [2, 26, 48, 51] *estimate* upper bounds for the unfairness across all downstream models and tasks based on the *training dataset*. However, there is *no guarantee* that these estimations give true upper bounds. These bounds can be *underestimated* because of overfitting to the training and validation sets. Thus, when their models are deployed on unseen test data, they can fail the desired fairness requirements. This calls for statistical guarantees such as high-confidence guarantees.

*High-confidence guarantees* are required to ensure that the unfairness across all downstream models and tasks will be *consistently* bounded with *high probabilities*. In many areas of supervised learning, providing high-confidence guarantees is considered essential for ensuring the fairness, privacy, and safety of the learning algorithm [1, 18, 43, 70]. This need becomes even more critical in the context of FRL as the absence of such guarantees can lead to undesired behaviors across multiple downstream applications. In FRL, a method called FARE [33] provides certificates that the downstream unfairness will be bound by some threshold with high probability. However, how to let users *explicitly control* the error thresholds and confidence levels jointly for the high-confidence guarantees is unexplored.

We propose *Fair Representation learning with high-confidence Guarantees* (FRG), the first work to our knowledge that guarantees with high confidence that the output representation models are "fair" according to *user-specified* thresholds of unfairness and confidence levels. The fairness criterion we focus on is demographic parity (DP) [19], a group fairness notion (the extension to equal opportunity and equalized odds will be discussed in Appendix C). A fair representation model ensures that, across arbitrary downstream tasks and models, the largest difference in positive prediction rate – denoted as $\Delta_{\text{DP}}$ (defined in Sec. 3) – is bounded by an error threshold $\varepsilon \in [0, 1]$. Thus, once a user designates a threshold $\varepsilon$ for $\Delta_{\text{DP}}$ and a confidence level, FRG guarantees with high *confidence* that any (including adversarial) downstream tasks and models using the representations generated by its learned representation model will not exceed the threshold $\varepsilon$. Equivalently, the representation models output by FRG will have a high probability of satisfying the desired fairness constraint even when applied to a worst-case adversarial downstream model.

FRG consists of three major components: (1) the *candidate selection* component proposes a representation model that will likely satisfy the fairness constraint with high probability; (2) the *adversarial inference* component aims to utilize the representations learned by the proposed model to adversarially predict the sensitive attributes to maximize $\Delta_{\text{DP}}$; and (3) the *fairness test* component establishes a high-confidence upper bound on the worst-case downstream $\Delta_{\text{DP}}$ based on the "optimal" adversarial prediction to determine whether the proposed model passes the test and should be returned.

We provide theoretical justification for the high-confidence fairness guarantees based on the assumption that the fairness test has access to an optimal adversary (defined in Sec. 5) which is approximated with optimization. We find a direct mapping between $\Delta_{\text{DP}}$ and the absolute covariance between the sensitive attributes and the predictions, so the optimal adversary can be achieved by maximizing the absolute covariance. Alternatively, it is possible to find a high-confidence upper bound via the upper bounds on $\Delta_{\text{DP}}$ as derived by previous work [26, 68] to avoid relying on training an optimal adversary. However, our study shows that these upper bounds are typically loose and thus impractical for establishing guarantees while preserving utility (Appendix M). Specifically, there exists a non-trivial gap between $\Delta_{\text{DP}}$ and its theoretical upper bound (as demonstrated in Appendix Figure 12 and 13).

In experiments, we use three real-world datasets each with 2-3 tasks to verify that FRG can indeed be used to learn fair representation models that satisfy the fairness criteria with the desired high probability. Compared to FRG, six state-of-the-art (SOTA) FRLs either violate the fairness constraints with non-trivial probability (at least $0.1$) or achieve lower predictive performance than FRG.

## 2   Related Work

Fair representation learning (FRL) has been studied for at least a decade [75]. While a stream of FRL studies optimizing the representations for a specific downstream task [10, 11, 24, 51, 62, 67, 74, 77]. numerous FRL methods learn general representations [2, 30, 52] that are fair, even when downstream

tasks are unknown or unlabeled. One category of these methods draws inspiration from information and probability theory [26, 31, 35, 37, 45, 54, 63, 65, 68, 72]. One work explores the use of distance covariance [44]. Some methods can limit downstream unfairness by constraining the total variation distance between the representation distributions of different groups [5, 48, 66, 76]. Other approaches promote independence from sensitive attributes through penalizing Maximum Mean Discrepancy [15, 47, 56] or statistical dependence [25, 61], meta-learning [57], PCA [39, 42], learning a shared feature space between groups [12], or disentanglement [14, 46, 55]. A stream of work uses adversarial training that limits the adversary's performance in predicting sensitive attributes [20, 22, 36, 60, 71]. Different from these methods, FRG constructs high-confidence guarantees based on a separately trained adversary without joint optimization with the primary objective under fairness constraints, which is considered more reliable than adversarial training.

Some FRLs provide theoretical analyses. Several works [26, 32, 48, 66, 76] prove upper bounds on the unfairness of all downstream models and tasks. These bounds can be estimated and verified with a training and validation set. However, these bounds may fail to generalize to an unseen test set. Some other works [5, 23, 33] provide statistical guarantees for test data. For example, FARE [33] provides practical certificates that serve as high-confidence upper bounds on downstream unfairness. Different from these methods, our framework provides an explicit way to *control both the confidence level* and *error threshold* for all downstream models, and yields tighter empirical bounds (Section 6).

Furthermore, in this study we focus on group fairness [19, 27], one of the most widely used fairness measures, including in legal setting, e.g., in the New York City Local Law 144 on Automated Employment Decision Tools and in the EEOC's rule of 80% hiring rates across sensitive groups. Some prior works [40, 41, 59, 64] focus on another important measure, i.e., individual fairness, without providing high confidence guarantees. Finally, some prior work [28, 43, 70] provides high-confidence guarantees for fair classification, but does not explore representation learning.

## 3   Preliminaries

We first introduce notations for representation learning and the unfairness measure we focus on. Let $X$ be a random variable denoting the *feature vector*, and $S$ a random variable denoting *sensitive attributes*. $D \coloneqq \{(X_i, S_i)\}_{i=1}^n$ denotes a dataset with i.i.d. data samples, where each $(X_i, S_i)$ has the same joint distribution as $(X, S)$. Let $\mathcal{D}$ be the set of all $D$'s, $\phi \in \Phi$ be the *representation model parameters*, and $q_\phi$ be the *representation model* parameterized by $\phi$. We define $Z$ as the *representation* for $(X, S)$ where $Z \sim q_\phi(\cdot | X, S)$ and $Z \in \mathbb{R}^l$.

The learned representation will be used for subsequent supervised learning *downstream* tasks. We denote the *label* for such a downstream task as the random variable $Y$. The objective in a downstream task is to predict $Y$ given $(X, S)$. It is common to use $Z$ in place of $(X, S)$ as input to a *downstream model* $\tau : \mathbb{R}^l \to \mathbb{R}$. Let $\hat{Y} \coloneqq \tau(Z)$ denote the prediction of $Y$ by model $\tau$. We call $\hat{Y}$ the *downstream prediction*. There can be multiple downstream tasks that make use of the same representation $Z$.

The goal is to learn a fair representation model that ensures a specified notion of fairness across downstream tasks and models. In this work, we focus on binary classification tasks[1] and a widely used group fairness objective called demographic parity (DP) [19]. The extension to Equal Oppertunity and Equalized Odds will be discussed in Appendix C. Below we formally define a measure of *demographic disparity* of how unfair a downstream model $\tau$ is under DP.

**Definition 3.1** (Demographic disparity measure). Let $\Delta_{\mathrm{DP}}(\tau, \phi)$ represent the measure of unfairness in the downstream predictions $\hat{Y}$ produced by model $\tau$ when using representation parameters $\phi$. Specifically, $\Delta_{\mathrm{DP}}(\tau, \phi) \coloneqq |\Pr(\hat{Y} = 1 | S = 1) - \Pr(\hat{Y} = 1 | S = 0)|$.

For simplicity, we assume that $Y$ and $S$ are binary, and this definition can be generalized to non-binary settings. When $S$ is non-binary, $\Delta_{\mathrm{DP}}(\tau, \phi)$ is defined as the maximum absolute difference between the conditional probabilities, $\Pr(\hat{Y} = 1 | S)$, with any pair of values of $S$ [8] (Appendix A).

## 4   Problem Formulation

This section formulates the task of representation learning with high-confidence fairness guarantees.

---

[1]FRG can be easily extended to provide similar guarantees for non-binary classification and regression by limiting $\mathrm{Cov}(S, Z)$. We focus on binary classification due to its prevalence in literature and legal systems.

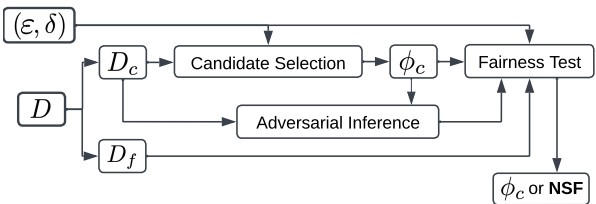

Figure 1: An overview of the FRG framework. Given a dataset $D$, with probability $1 - \delta$, FRG generates an "$\varepsilon$-fair" representation model, or returns NSF if such a model cannot be found. See Section 5 for discussion.

A fair representation model should ensure with high confidence that the representations it generates will not lead to unfairness for downstream tasks. Specifically, a representation model is fair if and only if it results in fair predictions (as defined in Def. 3.1) for every possible downstream model and downstream task. That is, for all downstream tasks and all $\tau$, $\Delta_{\text{DP}}(\tau, \phi)$ must be upper-bounded by a small constant, $\varepsilon$. We define an "$\varepsilon$-fair" representation model as follows.

**Definition 4.1** ("$\varepsilon$-fair" representation model). Representation model $q_\phi$ is $\varepsilon$-fair with parameter $\varepsilon \in [0, 1]$ if and only if $\Delta_{\text{DP}}(\tau, \phi) \leq \varepsilon$, for every downstream model $\tau$ and downstream task.

We define a representation learning algorithm $a : \mathcal{D} \to \Phi$ to be an algorithm that takes a data set as input and produces representation model parameters as output. In this paper, we aim to provide a representation learning algorithm such that any representation model it learns is guaranteed to be $\varepsilon$-fair under Def. 4.1, with high confidence. Such an algorithm has the following formal definition.

**Definition 4.2** (A representation learning algorithm with high-confidence fairness guarantees). Given $\varepsilon \in [0, 1], \delta \in (0, 1)$, and a dataset $D$, a representation learning algorithm $a$ is said to provide a $1 - \delta$ confidence $\varepsilon$-fairness guarantee if and only if $\Pr\left(g_\varepsilon(a(D)) \leq 0\right) \geq 1 - \delta$, where $g_\varepsilon(\phi) := \sup_\tau \Delta_{\text{DP}}(\tau, \phi) - \varepsilon$.

Observe that $q_\phi$ is an $\varepsilon$-fair representation model if and only if $g_\varepsilon(\phi) \leq 0$ (Def. 4.1). Therefore, any algorithm under Def. 4.2 guarantees that any representation model with parameters learned by this algorithm has at least $1 - \delta$ probability to be an $\varepsilon$-fair representation model. Algorithms of this form can generally be categorized as *Seldonian* algorithms [70]. This guarantee implies that even in the worst case (when downstream models are adversarial), any resulting representation model should *not* fail the $\varepsilon$-fairness constraint with probability larger than $\delta$.

**Special case: unachievable $\varepsilon$-fair representation models.** We note that in some scenarios, it may not be possible for any non-degenerate algorithm to ensure fairness with the specified confidence $1 - \delta$, e.g., when $\varepsilon, \delta$, and the amount of training data are all very small. In such cases, we allow the algorithm to output *No Solution Found* (NSF) as a way of indicating that it is unable to provide the required confidence that the learned representation will be fair given the amount of data it has been provided. To indicate that it is always fair for the algorithm to return NSF, we set $g_\varepsilon(\phi) = 0$. However, if an algorithm constantly returns NSF, it is of no value. We empirically evaluate the probability of returning a solution (i.e., not NSF) in Section 6.

## 5 Methodology

We now introduce our proposed framework, i.e., ***F**air **R**epresentation learning with high-confidence **G**uarantees* (FRG). It is the first representation learning algorithm that provides a *user-defined* high-confidence fairness guarantee. An overview of FRG is provided in Fig. 1.

FRG consists of three major components: *candidate selection*, *adversarial inference* and *fairness test*. First, we present a high-level summary of the algorithm before discussing each component in detail. FRG first splits the data $D$ into disjoint sets, $D_c$ and $D_f$. Candidate selection uses $D_c$ to optimize and propose *candidate solution $\phi_c$*. We train an adversarial model to predict sensitive attributes using the representations of $D_c$ as learned by the model $q_{\phi_c}$. The fairness test uses predictions made by the adversarial model on $D_f$ to evaluate whether $\phi_c$ can satisfy $g_\varepsilon(\phi_c) \leq 0$ on future unseen data with sufficient confidence. Finally, FRG returns $\phi_c$ if the fairness test is passed and NSF otherwise. While it is the fairness test that ensures the high-confidence guarantees, an effective candidate selection that reasons about the fairness test procedure is needed to maximize the likelihood of passing the test.

In this section, we first define an optimal adversary and propose an effective optimization for the adversarial model to achieve an approximation of the optimum. We then introduce the fairness

test assuming the access to an oracle optimal adversary. Lastly, we provide details for a candidate selection that proposes candidates that aim to pass the fairness test and achieve high expressiveness.

## 5.1 Adversarial Inference

To learn an adversarial downstream model $\tau_{\text{adv}}$ that best predicts the sensitive attribute $S$, we generate the representations for $D_c$ using a proposed candidate representation model $q_{\phi_c}$ as input to $\tau_{\text{adv}}$. We define an optimal adversary $\tau_{\text{adv}}^*$ to be one that maximizes the $\Delta_{\text{DP}}$ (Def. 5.1), and prove in Theorem 5.2 that when both $S$ and $\hat{Y}$ are binary, there exists a mapping between $\Delta_{\text{DP}}$ and the absolute covariance between $\hat{Y}$ and $S$ (denoted as $|\text{Cov}(\hat{Y}, S)|$). The extension to non-binary sensitive attributes is provided in Appendix D.

**Definition 5.1** (Optimal adversary). *Given a representation model $q_\phi$, a downstream model $\tau_{\text{adv}}^*$ is an optimal adversary if and only if $\tau_{\text{adv}}^* \in \arg\max_\tau \Delta_{\text{DP}}(\tau, \phi)$.*

**Theorem 5.2.** *Suppose $S, \hat{Y} \in \{0, 1\}$ are Bernoulli random variables. We have $\Delta_{\text{DP}}(\tau, \phi) = \frac{|\text{Cov}(\hat{Y}, S)|}{\text{Var}(S)}$.* **Proof.** *See Appendix B.*

Following Theorem 5.2, we have $\tau_{\text{adv}}^* \in \arg\max_\tau |\text{Cov}(\hat{Y}, S)| = \arg\max_\tau \Delta_{\text{DP}}(\tau, \phi)$. Intuitively, the worst-case $\Delta_{\text{DP}}$ is achieved when the adversary is optimal either in predicting $S$ or $1 - S$, and thus achieves either $\max_\tau \text{Cov}(\hat{Y}, S)$ or $\min_\tau \text{Cov}(\hat{Y}, S)$ (that are inversely correlated). In practice (Sec. 6), we find it sufficient to train an approximately optimal adversary to predict $S$ based on $Z$ using traditional gradient-based optimization strategies. We train $\tau_{\text{adv}}$ with representations and sensitive attributes from $D_c$, which will then be used by the fairness test to evaluate on $D_f$.

## 5.2 Fairness Test

The fairness test aims to evaluate whether a candidate solution $\phi_c$ induces a fair representation model with high confidence. In this section, we propose constructing a high-confidence upper bound on $g_\varepsilon(\phi)$, assuming access to an optimal adversary $\tau_{\text{adv}}^*$ (Def. 5.1), which in practice will be approximated by the adversarial model. If this high-confidence upper bound is at most zero, then we can conclude that $g_\varepsilon(\phi) \leq 0$ with confidence $1 - \delta$. We then detail the evaluation process for a candidate solution $\phi_c$ and show that it satisfies the $1 - \delta$ confidence $\varepsilon$-fairness guarantee.

### 5.2.1 $1 - \delta$ confidence upper bound on $g_\varepsilon(\phi)$

We define $U_\varepsilon : (\Phi, \mathcal{D}) \to \mathbb{R}$ to be such a function that produces a $1 - \delta$ confidence upper bound. Specifically, for $U_\varepsilon(\phi, D_f)$,

$$\Pr\Big(g_\varepsilon(\phi) \leq U_\varepsilon(\phi, D_f)\Big) \geq 1 - \delta. \tag{1}$$

The overall idea is to get unbiased estimates of $\Pr(\hat{Y} = y | S = s)$ of all combinations of $y$ and $s$, construct confidence intervals on these probabilities, and then compose these intervals to form the confidence upper bound on $g_\varepsilon$. It takes two different approaches to compute $U_\varepsilon$ when the sensitive attribute $S$ is binary v.s. multiclass due to their different definitions of $\Delta_{\text{DP}}$ (Def. 3.1 and Def. A.1). We will provide the approach for multi-class $S$ in Appendix F and focus on the binary case here.

We follow these steps. First, for each data point $(X_i, S_i) \in D_f$, we feed $Z_i = \phi(X_i, S_i)$ to the optimal adversary $\tau_{\text{adv}}^*$, which aims to infer $S$, and get output $\hat{Y}_i$. We separate $D_f$ into $D_{f,S=0}$ and $D_{f,S=1}$ such that all points in $D_{f,S=0}$ has $S = 0$ and all points in $D_{f,S=1}$ has $S = 1$. Each pair $((X_0^{(k)}, S_0^{(k)}), (X_1^{(k)}, S_1^{(k)}))$ can then be used to create a pair of unbiased estimates of $\Pr(\hat{Y} = 1 | S = s)$ where $s \in \{0, 1\}$, denoted as $\hat{p}^{(k)}(1|s)$. Thus, $m$ i.i.d. unbiased estimates of $\Pr(\hat{Y} = 1 | S = s)$ can be obtained by sampling $m$ pairs, i.e., $\mathbb{E}[\hat{p}^{(k)}(1|s)] = \Pr(\hat{Y} = 1 | S = s)$ for any $k \in [1, ..., m]$. By linearity of expectation [70], they form $m$ unbiased point estimates of $\Pr(\hat{Y} = 1 | S = 0) - \Pr(\hat{Y} = 1 | S = 1)$ which will be used to construct confidence intervals on $\Pr(\hat{Y} = 1 | S = 0) - \Pr(\hat{Y} = 1 | S = 1)$.

Second, we apply standard statistical tools such as Student's t-test [69], Hoeffding's inequality [29] etc., to construct a $1 - \delta$ confidence interval (CI) $[c_l, c_u]$ on $\Pr(\hat{Y} = 1 | S = 0) - \Pr(\hat{Y} = 1 | S = 1)$ using $\hat{p}^{(1)}(1|0) - \hat{p}^{(1)}(1|1), \ldots, \hat{p}^{(m)}(1|0) - \hat{p}^{(m)}(1|1)$. Finally, the $1 - \delta$ confidence upper bound of $g_\varepsilon = \sup_\tau |\Pr(\hat{Y} = 1 | S = 0) - \Pr(\hat{Y} = 1 | S = 1)| - \varepsilon$ can be obtained by taking $\max(|c_l|, |c_u|) - \varepsilon$.

Note that we use $\delta/2$ to obtain each of $c_l$ and $c_u$ to ensure that the bound on absolute value holds with probability at least $1 - \delta$ via the union bound.

While our framework is flexible to the techniques for achieving CIs, our experiments (Sec. 6) use the Student's t-test as it is well understood and used across the sciences for high-risk applications (e.g., biomedical research [50]). We include the procedure for constructing the confidence bounds with Student's t-test in Appendix G and more discussion on other variants of CIs in Appendix H.

### 5.2.2 Evaluation of candidate solutions

Suppose the fairness test gets a candidate solution $\phi_c$ and $U_\varepsilon(\phi_c, D_f) \leq 0$, it follows that there is at least confidence $1 - \delta$ that $g_\varepsilon(\phi_c) \leq 0$ (Inequality 1). Then, the fairness test concludes with at least $1 - \delta$ confidence that $q_{\phi_c}$ is an $\varepsilon$-fair representation model, and $\phi_c$ passes the test. If, however, $U_\varepsilon(\phi_c, D_f) > 0$, then the algorithm cannot conclude that $g_\varepsilon(\phi_c) \leq 0$ with high confidence. Therefore, the fairness test concludes that there is not sufficient confidence that $q_{\phi_c}$ is an $\varepsilon$-fair representation model, and $\phi_c$ fails the test.

Finally, if $\phi_c$ passes the fairness test, FRG outputs $\phi_c$. Otherwise, it outputs NSF. When $\phi_c$ fails, we do not search for and test another representation model because this would result in the well-known "multiple comparisons problem." In this case, each run of the fairness test can be viewed as a hypothesis test for determining whether the representation is fair with sufficient confidence.

### 5.2.3 Theoretical Analysis

In this section, we prove that the fairness test with access to an optimal adversary (Def. 5.1) provides FRG with the desired high confidence $\varepsilon$-fairness guarantee, i.e., the probability that it produces a representation that is not $\varepsilon$-fair for every downstream task and model is at most $\delta$.

**Theorem 5.3.** *Suppose fairness test finds $U_\varepsilon(\phi, D_f)$, a $1 - \delta$ confidence upper bound of $g_\varepsilon(\phi)$ for arbitrary $\phi$, then FRG provides a $1 - \delta$ confidence $\varepsilon$-fairness guarantee.* **Proof.** See Appendix E.

## 5.3 Candidate Selection

Candidate selection searches for a representation model using $D_c$ and proposes a candidate solution $\phi_c$ for the fairness test. Recall that the fairness test provides the desired $1 - \delta$ confidence $\varepsilon$-fairness guarantee (Def. 4.2) regardless of the choice of candidate selection (Theorem 5.3). However, candidate selection is considered ineffective if most of its proposed solutions fail the fairness test, which will lead to a high probability of returning NSF. In this section, we introduce an effective selection of candidates that are both likely to pass the fairness test and highly expressive.

### 5.3.1 Predicting Whether a Candidate Solution Will Pass the Fairness Test

Candidate selection proposes a candidate solution $\phi_c$ that it predicts will pass the fairness test. Such a prediction should leverage the knowledge of the exact form of the fairness test as much as possible, except using dataset $D_c$ instead of $D_f$, i.e., checking whether $U_\varepsilon(\phi_c, D_c) \leq 0$. There are two differences in practice because the candidate selection repeatedly searches for the candidate solution.

First, for efficiency, we cannot fully optimize for an adversarial downstream model from scratch for every candidate searched. Thus, after initializing an adversary $\hat{\tau}_{\mathrm{adv}}$, for each candidate searched, we take $t \in [1, 10]$ gradient steps (a hyperparameter) for optimizing $\hat{\tau}_{\mathrm{adv}}$, without reinitialization.

Second, we repeatedly use the same dataset $D_c$ to construct high confidence upper bounds and thus, may overfit to $D_c$, resulting in an overestimation of the confidence that the candidate solution will pass the fairness test and causing more NSF. One way to mitigate this issue is to inflate the confidence interval used in candidate selection. We multiply the confidence upper bound by $\alpha$ where $\alpha \geq 1$ is a hyperparameter, i.e., $\hat{U}_\varepsilon(\phi_c, D_c) := \alpha U_\varepsilon(\phi_c, D_c)$ (Appendix L provides a case to show why such inflation could be critical in reducing the chance of getting NSF). We find $\hat{U}_\varepsilon(\phi_c, D_c)$, the inflated $1 - \delta$ confidence upper bound on $g_\varepsilon(\phi_c)$, following a similar procedure as Sec. 5.2.1, and use the constraint $\hat{U}_\varepsilon(\phi_c, D_c) \leq 0$ to find a candidate solution that reasons about the fairness test to increase the likelihood of passing.

### 5.3.2 Optimizing for a Candidate Solution With a Constrained Objective

In addition to candidate solutions that are likely to pass the fairness test, candidate selection also favors solutions that have high expressiveness, so that the representations they generate are effective for downstream tasks. We achieve this without being limited to a specific learning algorithm. We support most parameterized representation learning architectures proposed in previous work, including the

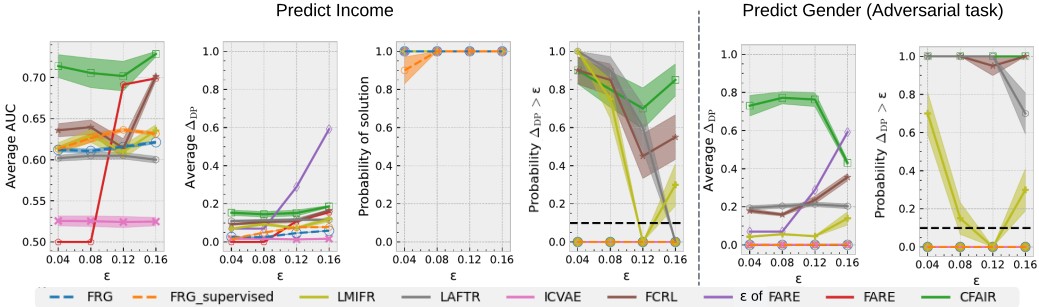

Figure 2: The evaluation on the **Adult** dataset. The target label is *income* and the sensitive attribute is *gender*. We vary $\varepsilon \in \{0.04, 0.08, 0.12, 0.16\}$. $\delta$ is fixed at 0.1. The four plots on the left are: (1) the average AUC; (2) the average $\Delta_{\text{DP}}$; (3) the fraction of trials that returns a solution excluding NSF; (4) the fraction of trials that violates $\Delta_{\text{DP}}(\tau, \phi) \le \varepsilon$ on the ground truth dataset. The AUC and $\Delta_{\text{DP}}$ on the adversarial task are on the right.

VAE-based methods [38, 47], contrastive learning methods [26, 55], etc. In our experiments, we focus on an adaptation of VAE [47] to construct the objective function that candidate selection optimizes. Specifically, we define $X \sim p_\theta(\cdot | Z, S)$ as the generative model for $X$ with input $(Z, S)$, parameterized by $\theta$. Let $\mathbb{KL}$ denote KL-divergence, and $p(Z)$ be a standard isotropic Gaussian prior, i.e., $p(Z) = \mathcal{N}(0, \mathbf{I})$, where $\mathbf{I}$ is the identity matrix. Overall, we define the candidate selection process as approximating a solution to the constrained optimization problem:

$$\max_{\theta, \phi} \; \mathbb{E}_{q_\phi(Z|X,S)}\Big[ \log p_\theta(X|Z,S) \Big] - \mathbb{KL}\Big( q_\phi(Z|X,S) \| p(Z) \Big) \qquad \text{s.t.} \quad \hat{U}_\varepsilon(\phi, D_c) \le 0. \quad (2)$$

We propose using a gradient-based optimization to approximate an optimal solution $(\theta, \phi)$. When gradient-based optimizers are used, the inequality constraint can be incorporated into the objective using the KKT conditions. That is, we find saddle-points of the following Lagrangian function:

$$\mathcal{L}(\theta, \phi; \lambda) := -\mathbb{E}_{q_\phi(Z|X,S)}\Big[ \log p_\theta(X|Z,S) \Big] + \mathbb{KL}\Big( q_\phi(Z|X,S) \| p(Z) \Big) + \lambda \hat{U}_\varepsilon(\phi, D_c), \quad (3)$$

where $\lambda \ge 0$ is a learnable Lagrange multiplier. Note that after each gradient step in the optimization, we need to update the adversary $\hat{\tau}_{\text{adv}}$ as mentioned above before evaluating $\hat{U}_\varepsilon(\phi, D_c)$.

This candidate selection procedure does not require any supervision. However, if a downstream task with labels is given, a supervised loss (e.g., binary cross-entropy) can be applied to $\mathcal{L}(\theta, \phi; \lambda)$ to improve the downstream predictions. We evaluate FRG both with and without supervision in Sec. 6.

## 6 Experiments

Here, we evaluate the performance and fairness of FRG, focusing on the following research questions. **RQ1:** Do the empirical results align with our expectation that FRG produces $\varepsilon$-fair representation models with high confidence? In other words, is $\Delta_{\text{DP}}$ of all downstream models and tasks upper-bounded by a desired $\varepsilon$ with high probability? To address this question, we estimate the probabilities of violating the constraint $\Delta_{\text{DP}}(\tau, \phi) \le \varepsilon$ using results from multiple runs of the algorithm with different training sets. **RQ2:** Can FRG learn expressive representations that are useful for downstream predictions? We evaluate the prediction performance on datasets with 2-3 downstream tasks using the area under the ROC curve (AUC), and compare its values across methods achieving similar demographic disparity bounds. **RQ3:** Would FRG frequently result in NSF to avoid unfairness even with sufficient data and reasonable values of $\varepsilon$ and $\delta$ due to an ineffective candidate selection? To address this question, we evaluate the probability that FRG provides a solution other than NSF.

### 6.1 Experiment Setup

**Datasets.** We use three real-world datasets each with at least two downstream tasks, including the adversarial tasks that predict the sensitive attributes: UCI *Adult* [6] and *Income* (California only, commonly known as Retiring Adult) [17] both with *2* downstream tasks, and Heritage *Health* [34] with *3* downstream tasks. All downstream tasks and sensitive attributes are listed per dataset in Appendix Table 1, including their basic statistics (see details in Appendix I). For each dataset, we use the first downstream task for hyperparameter search and validation, and the last task is *adversarial task* predicting the sensitive attribute.

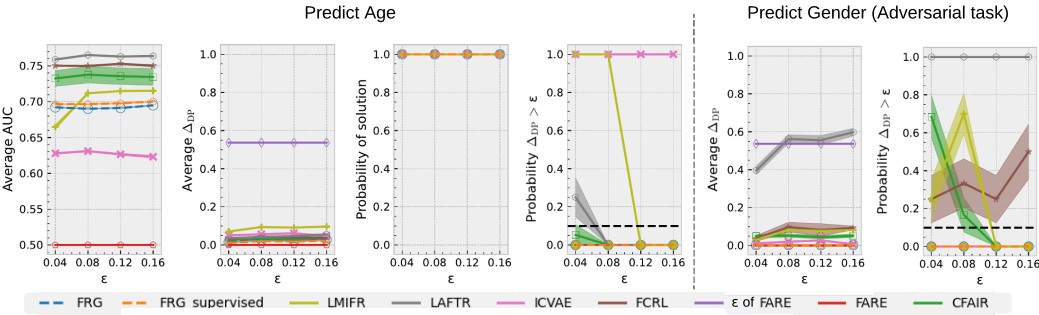

Figure 3: The evaluation on the **Health** dataset with sensitive attribute *gender*. The original target task predicting *Charlson Index* is in Appendix Figure 5. Here, we show the *transfer learning* capabilities on predicting *age*.

**Baselines.** We consider six competitive FRL baselines. *LAFTR* [48] proposes limiting the unfairness of arbitrary downstream classifiers with adversarial training. *ICVAE* [54] uses an upper-bound of the mutual information between $S$ and $\hat{Y}$ as a regularizer. *LMIFR* [68] uses Lagrangian Multipliers to encourage a representation model to satisfy constraints that upper-bound the mutual information between $S$ and $\hat{Y}$. *CFAIR* [76] adopts a balanced error rate (BER) and conditional learning to achieve parity. *FCRL* [26] proposes controlling parity via contrastive information estimators. *FARE* [33] provides high-confidence certificates with representations drawn from discrete distributions with finite support for downstream unfairness. All methods except FARE estimate upper bounds of $\Delta_{\mathrm{DP}}$ on arbitrary downstream models using training data, which may not generalize to unknown test data.

For the Adult and Health datasets we train *LAFTR*, *FCRL*, *CFAIR*, and *FARE* in a supervised manner using the first downstream tasks (Appendix Table 1) because their model architectures rely on a supervised downstream task and avoiding it causes large performance decreases. *CVIB*, *LMIFR* are unsupervisedly trained without a labeled downstream task. For these two datasets, we train FRG with supervision (denoted as *FRG_supervised*) and without supervision (denoted as *FRG*). For the Income dataset, all models are trained with a supervised loss because the task is difficult for all models.

**Evaluation process.** For each dataset, we split the data into training ($D_{\mathrm{train}}$), validation ($D_{\mathrm{val}}$), and test ($D_{\mathrm{test}}$) sets according to ratio 0.6:0.2:0.2. For FRG, we sample 10% of the training set to be $D_f$ for fairness test and let candidate selection use the remaining 90% as $D_c$. We run experiments across 4-5 different $\varepsilon$'s. We fix $\delta = 0.1$ for the main experiment, i.e., we want the probability of violating the constraint to be at most 0.1 and provide additional study on various $\delta$'s in Appendix Figure 8. In one experiment, we train all methods *20 times* with different randomly drawn training sets to get 20 representation models, which will then be applied to all downstream tasks. We report averages over the 20 trials for AUC, $\Delta_{\mathrm{DP}}$, the probability of returning a solution, and the probability of failing the constraint $\Delta_{\mathrm{DP}}(\tau, \phi) \leq \varepsilon$. All figures plot the error bars evaluated with standard deviations.

**Hyperparameter tuning.** The goal of hyperparameter tuning is to find a set of parameters that achieve high downstream performance while satisfying $\Delta_{\mathrm{DP}}(\tau, \phi) \leq \varepsilon$. Thus, we use validation sets of the *first* downstream tasks (Appendix Table 1) for tuning. We repeat the training for each parameter set at least 3 times. The first evaluation criterion is whether the constraint $\Delta_{\mathrm{DP}}(\tau, \phi) \leq \varepsilon$ is satisfied. If finding a set of parameters that satisfies the constraint is impossible, we select the one that achieves the smallest $\Delta_{\mathrm{DP}}$. For FRG, we also prioritize the parameters that achieve the lowest probability of returning NSF. If there are ties, we choose the parameter set that achieves the highest average AUC. We note that the architectures and hyperparameters for the downstream models are consistent across methods for fair comparison. More details are provided in Appendix J.

## 6.2 Result and Discussion

The experiment results for the three datasets are provided in Figures 2, 3, and 4. For Health, the evaluation of the targeted downstream task is provided in Appendix Figure 5.

Overall, both FRG and FRG_supervised can maintain $\Delta_{\mathrm{DP}} \leq \varepsilon$ with a sufficiently high probability (at least 0.9). In contrast, most baseline methods cannot consistently satisfy $\varepsilon$-fairness with high probability across all datasets. For baseline methods, we observe that a smaller $\varepsilon$ causes a larger probability of failing the constraint. They also tend to fail on the adversarial tasks, in contrast to FRG (rightmost panels in Figures 2, 3, and 4). Thus, FRG provides high-confidence fairness guarantees across tasks for different $\varepsilon$'s (**RQ1**). To answer **RQ2**, we highlight that FRG can match or outperform

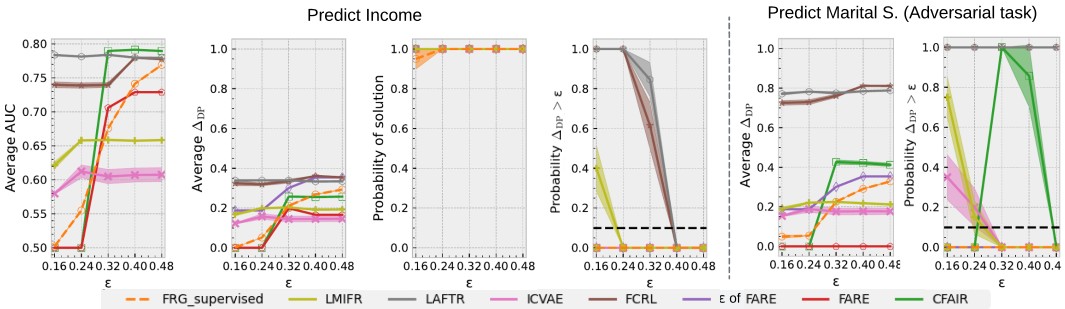

Figure 4: The evaluation on the **Income** dataset. The target label is *income* and the sensitive attribute is *marital status* which has 5 classes. We vary $\varepsilon \in \{0.16, 0.24, 0.32, 0.40, 0.48\}$.

baselines in terms of prediction performance (AUC). Compared with baselines that achieve $\Delta_{\mathbf{DP}} \leq \varepsilon$ with high probability (*ICVAE* for Adult and Health, and *FARE*), FRG tends to yield higher AUC, especially when $\varepsilon$ is small. Further comparisons of the tradeoff between AUC and $\Delta_{\mathbf{DP}}$ in Appendix K confirm that FRG tends to achieve the highest AUC among the methods that achieve $\Delta_{\mathbf{DP}} \leq \varepsilon$ with high probability. Furthermore, FRG also keeps a high probability of returning a solution (at least 0.9) over all datasets, which demonstrates the effectiveness of our candidate selection and addresses **RQ3**.

**Comparison between FARE and FRG.** Similar to FRG, FARE can also achieve high probability of satisfying the fairness constraint across all datasets. While FARE does not support user-defined $\varepsilon$'s, we use hyperparameter search (as discussed) to manipulate its high probability upper-bound of $\Delta_{\mathrm{DP}}$. However, these certificates are loose and often several times larger than the desired $\varepsilon$ (compare the "$\varepsilon$ of FARE" with the $x$-axes in Figures 2, 3, and 4). Additionally, even through this hyperparameter search, FARE does not certify fairness with enough granularity, i.e., the same certificates are given to multiple $\varepsilon$'s. Perhaps FARE's use of representations drawn from a discrete distribution with a finite support limits the representations' variability. In contrast, FRG provides the high confidence guarantees and achieves accurate downstream models even for small user-specified $\varepsilon$'s.

**On the primary downstream tasks (left Figures 2, 4, and 5).** All methods use the first tasks in Table 1 as the *target task* for hyperparameter search such that the models do not violate the fairness constraints on validation sets. Even though baseline methods including LMIFR, LAFTR, FCRL and CFAIR can satisfy the constraints on the validation set, they still fail with a large probability on the test set for the same task. In contrast, FRG and FARE can still keep $\Delta_{\mathbf{DP}}$ low with a high probability. This suggests that their theoretical upper bounds estimated using the training dataset overfit. Even with a hold-out validation set, it is still possible to underestimate the $\Delta_{\mathbf{DP}}$ for new test datasets. In comparison, FRG's high-confidence guarantees include statistical testing with held-out data, $D_f$, which automatically tests for and avoids unfair models resulting from overfitting.

**On the adversarial tasks (right Figures 2, 3, and 4).** Here, we check whether the learned representations can be used to adversarially infer sensitive attributes. Compared to other downstream tasks, the failure rates have increased for baseline methods (including LMIFR, LAFTR, FCRL, ICVAE, CFAIR), especially when $\varepsilon$ is small. Even though these methods provide an estimated upper bound for $\Delta_{\mathrm{DP}}$ in the worst case, the fairness constraints are still violated on these adversarial tasks due to the lack of high-confidence guarantees. However, FRG and FARE satisfy the constraints for unknown adversarial tasks, and FRG provides high-confidence *user-defined* fairness guarantees.

**On transfer learning (left Figure 3).** We first note that we use unsupervised learning for FRG, LMIFR and ICVAE on the Adult and Health datasets. So their performances on income prediction for Adult (left Fig. 2) and Charlson Index prediction (Appendix Fig. 5) for Health can also be used to demonstrate their transferability. Here, we focus on left Fig. 3 where task-specific labels are not exposed during training to all methods. Several baselines (LMIFR, ICVAE and LAFTR) increase their probability of violating the constraints compared to their performance on the target task that predicts Charlson Index. This may be the effect of overfitting the fairness constraint to a specific downstream task while not generalizing to all tasks. When the task is different, the sensitive information in the same representations can be exploited. Most supervised methods yield the lowest AUC scores (FARE, LAFTR, CFAIR). This may suggest that transferring the representations to a different task can hurt the prediction for supervised learning approaches.

**Supervised v.s. unsupervised FRG.** On the Adult and the Health datasets, although the supervised FRG performs slightly better than the unsupervised one, the improvement is insignificant. We hypothesize that while supervision helps, a more predictive candidate can also expose more sensitive information, making it easier to violate the fairness constraints. When the candidate selection aims to control $\hat{U}_\varepsilon(\phi, D_c) \leq 0$, the better-performing candidate may not be selected. In some cases when the candidate selection returns the better performing candidate, it can still fail the fairness test and be replaced by NSF if $U_\varepsilon(\phi_c, D_f) > 0$ (e.g., on the Adult dataset when $\varepsilon$ is small).

We further study the effect of different confidence levels $\delta \in \{0.01, 0.05, 0.1, 0.15\}$ (Appendix Figure 8). The performance and the $\Delta_{\mathrm{DP}}$ are similar when $\varepsilon$ is small on both the target and adversarial tasks. On the target task, as $\varepsilon$ gets large, e.g., $\varepsilon = 0.16$, the performance for the larger $\delta$ is marginally better than for the small $\delta$ while the $\Delta_{\mathrm{DP}}$'s are still similar. Overall, by increasing $\delta$, one might gain a marginal improvement in the prediction accuracy or performance but at the cost of reducing the confidence in the fairness guarantees. Finally, we study varying $\alpha$'s in Appendix L. Other evaluation metrics including F1, Average Accuracy, Equal Opportunity Difference, Equalized Odds Difference are provided in Appendix Figures 9, 10, and 11.

## 7 Conclusion and Limitations

In this work, we introduced FRG, an FRL framework that provides high-confidence fairness guarantees, ensuring that demographic disparity for all downstream models and tasks is upper-bounded by a *user-defined* error threshold and confidence level. Our work is substantiated with theoretical analysis, and our empirical evaluation demonstrates FRG's effectiveness across various downstream tasks.

The theoretical guarantees of FRG make several assumptions. First, we assume all data samples are i.i.d. Second, the use of Student's t-test assumes the point estimates of $g$ are normally distributed, which requires a large sample such that CLT holds. Third, we assume access to an optimal adversary (Def. 5.1) that uses representations as input to predict the sensitive attributes to maximize $\Delta_{\mathrm{DP}}$. We approximate it with an independently trained model.

In the future, FRG can be extended to provide guarantees related to measures of fairness, privacy [18], safety, robustness or concept erasure [7]. While FRG can account for label shift and concept drift because our guaranteed constraint on demographic disparity does not require any assumptions about the distribution of the downstream labels, future work could study guarantees under distributional shifts in features $X$ and/or sensitive attributes $S$. Future work could also consider other approaches without assuming access to an optimal adversary (Appendix M).

## 8 Acknowledgments

This work is supported by the National Science Foundation under grant no. CCF-2018372, by a gift from the Berkeley Existential Risk Initiative, and by Rutgers SAS Research Grant in Academic Themes. Philip S. Thomas and Przemyslaw A. Grabowicz took similar advising roles on this project. The authors would also like to thank Linjun Zhang at Rutgers for the time to review and discuss this work.

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

# A $\Delta_{\text{DP}}$ for Multi-class Sensitive Attributes

**Definition A.1** ($\Delta_{\text{DP}}$ for multiclass $S$). We define $\Delta_{\text{DP}}$ for multi-class sensitive attributes $S \in \mathcal{S}$ where $|\mathcal{S}| > 2$ as follows.

$$\Delta_{\text{DP}}(\tau, \phi) := \max_{i,j \in \mathcal{S}} \left| \Pr(\hat{Y} = 1|S = i) - \Pr(\hat{Y} = 1|S = j) \right|. \tag{4}$$

It follows from the implementation by Bird et al. [8].

# B Proof of Theorem 5.2

To prove $\Delta_{\text{DP}}(\tau, \phi) = \frac{|\text{Cov}(S,\hat{Y})|}{\text{Var}(S)}$, we first prove the following lemma. To simplify the notations in the proofs, we define $p_{a,b} := \Pr[\hat{Y} = a, S = b]$ where $a, b \in \{0, 1\}$.

**Lemma B.1.** *Suppose $S, \hat{Y} \in \{0, 1\}$ are Bernoulli random variables.*

$$\text{Cov}(\hat{Y}, S) = \Pr[\hat{Y} = 1, S = 1] \Pr[\hat{Y} = 0, S = 0] - \Pr[\hat{Y} = 1, S = 0] \Pr[\hat{Y} = 0, S = 1] \tag{5}$$

$$= p_{1,1} p_{0,0} - p_{1,0} p_{0,1}. \tag{6}$$

**Proof.**

$$\text{Cov}(\hat{Y}, S) = \mathbb{E}[\hat{Y}, S] - \mathbb{E}[\hat{Y}]\mathbb{E}[S] \qquad \text{(by definition of covariance)} \tag{7}$$

$$= \sum_{\hat{y},s} p_{\hat{y},s} \cdot (\hat{y} \cdot s) - \Pr[\hat{Y} = 1] \Pr[S = 1] \tag{8}$$

$$= p_{1,1} - \Pr[\hat{Y} = 1] \Pr[S = 1] \tag{9}$$

$$= p_{1,1} - (p_{1,1} + p_{1,0})(p_{1,1} + p_{0,1}) \tag{10}$$

$$= p_{1,1} - p_{1,1} p_{0,1} - p_{1,1} p_{1,0} - p_{1,1} p_{1,1} - p_{1,0} p_{0,1} \tag{11}$$

$$= p_{1,1}(1 - p_{0,1} - p_{1,0} - p_{1,1}) - p_{1,0} p_{0,1} \tag{12}$$

$$= p_{1,1} p_{0,0} - p_{1,0} p_{0,1} \tag{13}$$

Completing the proof.

**Theorem B.2** (Theorem 5.2 restated). *Suppose $S, \hat{Y} \in \{0, 1\}$ are Bernoulli random variables. Then $\Delta_{\text{DP}}(\tau, \phi) = \frac{|\text{Cov}(\hat{Y},S)|}{\text{Var}(S)}$.*

**Proof.**

$$\Delta_{\text{DP}}(\tau, \phi) = \left| \Pr(\hat{Y} = 1|S = 1) - \Pr(\hat{Y} = 1|S = 0) \right| \qquad \text{(By Def. 3.1)} \tag{14}$$

$$= \left| \frac{p_{1,1}}{\Pr(S = 1)} - \frac{p_{1,0}}{\Pr(S = 0)} \right| \tag{15}$$

$$= \left| \frac{p_{1,1} \Pr(S = 0) - p_{1,0} \Pr(S = 1)}{\Pr(S = 1) \Pr(S = 0)} \right| \tag{16}$$

$$= \frac{1}{\text{Var}(S)} \left| p_{1,1} p_{0,0} + p_{1,1} p_{1,0} - p_{1,0} p_{0,1} - p_{1,0} p_{1,1} \right| \tag{17}$$

$$= \frac{1}{\text{Var}(S)} \left| p_{1,1} p_{0,0} - p_{1,0} p_{0,1} \right| \tag{18}$$

$$= \frac{|\text{Cov}(\hat{Y}, S)|}{\text{Var}(S)} \qquad \text{(By Lemma B.1)} \tag{19}$$

Completing the proof.

## C  Extension to Equal Opportunity and Equalized Odds

In this section, we demonstrate that FRG can be extended to other group fairness metrics beyond demographic disparity ($\Delta_{\mathrm{DP}}$), specifically, Equal Opportunity Difference ($\Delta_{\mathrm{EOP}}$) and Equalized Odds Difference ($\Delta_{\mathrm{EOD}}$), with the assumption that the downstream task's labels are available. We start with their formal definitions.

**Definition C.1** (Equal Opportunity Difference).

$$\Delta_{\mathrm{EOP}}(\tau, \phi) := |\Pr(\hat{Y} = 1 | S = 1, Y = 1) - \Pr(\hat{Y} = 1 | S = 0, Y = 1)| \tag{20}$$

$$= \Delta_{\mathrm{DP}}(\tau, \phi | Y = 1). \tag{21}$$

**Definition C.2** (Equalized Odds Difference).

$$\Delta_{\mathrm{EOD}}(\tau, \phi) := \max_{y \in \{0,1\}} \left( \left| \Pr(\hat{Y} = 1 | S = 1, Y = y) - \Pr(\hat{Y} = 1 | S = 0, Y = y) \right| \right) \tag{22}$$

$$= \max \left( \Delta_{\mathrm{DP}}(\tau, \phi | Y = 1), \Delta_{\mathrm{DP}}(\tau, \phi | Y = 0) \right). \tag{23}$$

Here $\Delta_{\mathrm{DP}}(\tau, \phi | Y = y)$ for $y \in \{0, 1\}$ represents the $\Delta_{\mathrm{DP}}$ under the conditional distribution $(X, S | Y = y)$. Empirically, it is the $\Delta_{\mathrm{DP}}$ evaluated on the data samples whose downstream labels satisfy $Y = y$.

Thus, following the same procedure introduced in Sec. 5 on the data samples whose $Y = 1$, FRG can produce representation models that satisfy $\Delta_{\mathrm{EOD}}(\tau, \phi) = \Delta_{\mathrm{DP}}(\tau, \phi | Y = 1) \leq \varepsilon$ with probability at least $(1 - \delta)$.

By splitting $\delta$ in half, FRG (Sec. 5) can generate a representation model that satisfies $\Delta_{\mathrm{DP}}(\tau, \phi | Y = 1) \leq \varepsilon$ and $\Delta_{\mathrm{DP}}(\tau, \phi | Y = 0) \leq \varepsilon$, each with probability at least $(1 - \delta/2)$. By union bound, such a representation model satisfies $\Delta_{\mathrm{EOD}}(\tau, \phi) = \max \left( \Delta_{\mathrm{DP}}(\tau, \phi | Y = 1), \Delta_{\mathrm{DP}}(\tau, \phi | Y = 0) \right) \leq \varepsilon$ with probability at least $(1 - \delta)$.

## D  The relationship between $\Delta_{\mathrm{DP}}$ and $|\mathrm{Cov}(\hat{Y}, S)|$ for non-binary sensitive attributes

The standard definition of covariance is not applicable to non-binary categorical random variables like the sensitive attributes. The reason is that the covariance takes the numerical values of the random variable into account but the numerical values of the sensitive attribute has no actual meaning. However, we can define auxiliary random variables for $S$ for each pair of sensitive categories $i, j \in \mathcal{S}$, to represent $S$ as a set of binary indicator variables, such that covariance can be applied. This approach enables the application of FRG to the setting of non-binary sensitive attributes $S$.

Suppose $S \in \mathcal{S}$ where $|\mathcal{S}| > 2$. Create one indicator variable $S'_{i,j}$ for each pair of $i, j \in \mathcal{S}$ where $i \neq j$ such that $S'_{i,j} = 0$ if $S = i$ and $S'_{i,j} = 1$ if $S = j$. We denote $p_{a,b} := \Pr[\hat{Y} = a, S = b]$ where $a \in \{0, 1\}$ and $b \in \mathcal{S}$. We will first prove the lemma below before stating the main theorem.

**Lemma D.1.** $\mathrm{Cov}(\hat{Y}, S'_{i,j} | S \in \{i, j\}) = \frac{p_{1,j} p_{0,i} - p_{1,i} p_{0,j}}{(\Pr(S=i) + \Pr(S=j))^2}$.

**Proof.** Following the same proof as Lemma B.1 in Appendix B, we have

$$\mathrm{Cov}(\hat{Y}, S'_{i,j} | S \in \{i, j\}) \tag{24}$$

$$= \Pr[\hat{Y} = 1, S'_{i,j} = 1 | S \in \{i, j\}] \Pr[\hat{Y} = 0, S'_{i,j} = 0 | S \in \{i, j\}] \tag{25}$$

$$- \Pr[\hat{Y} = 1, S'_{i,j} = 0 | S \in \{i, j\}] \Pr[\hat{Y} = 0, S'_{i,j} = 1 | S \in \{i, j\}] \tag{26}$$

$$= \Pr[\hat{Y} = 1, S = j | S \in \{i, j\}] \Pr[\hat{Y} = 0, S = i | S \in \{i, j\}] \tag{27}$$

$$- \Pr[\hat{Y} = 1, S = i | S \in \{i, j\}] \Pr[\hat{Y} = 0, S = j | S \in \{i, j\}]. \tag{28}$$

Since $\Pr[S \in \{i, j\}] = \Pr(S = i) + \Pr(S = j)$, $\mathrm{Cov}(\hat{Y}, S'_{i,j} | S \in \{i, j\}) = \frac{p_{1,j} p_{0,i} - p_{1,i} p_{0,j}}{(\Pr(S=i) + \Pr(S=j))^2}$.

$\square$

Demographic disparity can be defined separately for each pair of sensitive categories, $i, j \in \mathcal{S}$, as $\Delta_{\text{DP}}^{i,j} = \left| \Pr(\hat{Y} = 1 | S = i) - \Pr(\hat{Y} = 1 | S = j) \right|$. Then, we can limit $\Delta_{\text{DP}} = \max_{i,j} \Delta_{\text{DP}}^{i,j}$, to ensure demographic parity with respect to any pair of sensitive categories [8]. Next, we provide the relationship between $\Delta_{\text{DP}}$ and $\text{Cov}(\hat{Y}, S'_{i,j} | S \in \{i, j\})$.

**Theorem D.2.**

$$\Delta_{\text{DP}}(\tau, \phi) = \max_{i,j} \left( 2 + \frac{\Pr(S = i)}{\Pr(S = j)} + \frac{\Pr(S = j)}{\Pr(S = i)} \right) \left| \text{Cov}(\hat{Y}, S'_{i,j} | S \in \{i, j\}) \right|,$$

where $i, j \in \mathcal{S}$ and $i \neq j$.

**Proof.**

$$\Delta_{\text{DP}}(\tau, \phi) = \max_{i,j} \left| \Pr(\hat{Y} = 1 | S = i) - \Pr(\hat{Y} = 1 | S = j) \right| \tag{29}$$

$$= \max_{i,j} \left| \frac{\Pr(\hat{Y} = 1, S = i)}{\Pr(S = i)} - \frac{\Pr(\hat{Y} = 1, S = j)}{\Pr(S = j)} \right| \tag{30}$$

$$= \max_{i,j} \left| \frac{\Pr(\hat{Y} = 1, S = i) \Pr(S = j) - \Pr(\hat{Y} = 1, S = j) \Pr(S = i)}{\Pr(S = i) \Pr(S = j)} \right| \tag{31}$$

$$= \max_{i,j} \frac{1}{\Pr(S = i) \Pr(S = j)} \left| p_{1,i} p_{0,j} + p_{1,i} p_{1,j} - p_{1,j} p_{0,i} - p_{1,j} p_{1,i} \right| \tag{32}$$

$$= \max_{i,j} \frac{1}{\Pr(S = i) \Pr(S = j)} \left| p_{1,i} p_{0,j} - p_{1,j} p_{0,i} \right| \tag{33}$$

$$= \max_{i,j} \frac{(\Pr(S = i) + \Pr(S = j))^2}{\Pr(S = i) \Pr(S = j)} \left| \frac{p_{1,i} p_{0,j} - p_{1,j} p_{0,i}}{(\Pr(S = i) + \Pr(S = j))^2} \right| \tag{34}$$

$$= \max_{i,j} \left( 2 + \frac{\Pr(S = i)}{\Pr(S = j)} + \frac{\Pr(S = j)}{\Pr(S = i)} \right) \left| \text{Cov}(\hat{Y}, S'_{i,j} | S \in \{i, j\}) \right| \tag{35}$$

This completes the proof. Using this relationship, the optimal adversary can be approximated for non-binary sensitive features.

For instance, suppose that $\Pr(S = i) = \frac{1}{|\mathcal{S}|}$ for all $i$. Then, $\Delta_{\text{DP}}$ is minimized when the predicted label $\hat{Y}$ does not provide any information differentiating any pair of $i$ and $j \in \mathcal{S}$. On the other hand, $\Delta_{\text{DP}}$ is maximized if there exists a pair of $i$ and $j$ such that $\hat{Y}$ is maximally correlated with $S$ conditioning on $S \in \{i, j\}$.

# E    Proof of Theorem 5.3

**Theorem E.1** (Theorem 5.3 restated). *Suppose fairness test finds $U_\varepsilon(\phi, D_f)$, a $1 - \delta$ confidence upper bound of $g_\varepsilon(\phi)$ for arbitrary $\phi$, then FRG provides a $1 - \delta$ confidence $\varepsilon$-fairness guarantee.*

**Proof.** By Def. 4.2, if FRG satisfies $\Pr\left(g_\varepsilon(a(D)) \leq 0\right) \geq 1 - \delta$, then FRG provides the desired $1 - \delta$ confidence $\varepsilon$-fairness guarantee. We prove by contradiction that $\Pr\left(g_\varepsilon(a(D)) \leq 0\right) \geq 1 - \delta$ if $a$ corresponds to FRG and $a(D)$ corresponds to the representation model parameters returned by FRG when run on dataset $D$.

We begin by assuming the result is false and then derive a contradiction. The beginning assumption is that $\Pr\left(g_\varepsilon(a(D)) \leq 0\right) < 1 - \delta$. By contrapositive, we have $\Pr\left(g_\varepsilon(a(D)) > 0\right) \geq \delta$. By the construction of FRG, $a(D)$ is either NSF or the proposed candidate solution $\phi_c \in \Phi$. Notice that $g_\varepsilon(a(D)) > 0$ if and only if $a(D)$ does not return NSF but returns $\phi_c$ instead, i.e., $a(D) = \phi_c$. The fairness test in FRG returns $\phi_c$ if and only if $U_\varepsilon(\phi_c, D_f) \leq 0$ (Sec. 5.2). Therefore, the following events are equivalent ($\Pr(A, B)$ denotes the joint probability of $A$ and $B$):

$$(g_\varepsilon(a(D)) > 0) \tag{36}$$

$$\iff (g_\varepsilon(a(D)) > 0, a(D) = \phi_c, U_\varepsilon(\phi_c, D_f) \leq 0) \tag{37}$$

$$\iff (g_\varepsilon(\phi_c) > U_\varepsilon(\phi_c, D_f), a(D) = \phi_c). \tag{38}$$

The joint event $(g_\varepsilon(\phi_c) > U_\varepsilon(\phi_c, D_f), a(D) = \phi_c)$ implies $(g_\varepsilon(\phi_c) > U_\varepsilon(\phi_c, D_f))$. Therefore,

$$\Pr\left(g_\varepsilon(\phi_c) > U_\varepsilon(\phi_c, D_f)\right) \geq \Pr\left(g_\varepsilon(\phi_c) > U_\varepsilon(\phi_c, D_f), a(D) = \phi_c\right) \geq \delta.$$

However, by construction of the fairness test and assuming the evaluated high confidence upper bound $U_\varepsilon(\phi_c, D_f)$ is correct, $\Pr(g_\varepsilon(\phi_c) \leq U_\varepsilon(\phi_c, D_f)) \geq 1 - \delta$ (Inequality 1), which implies $\Pr(g_\varepsilon(\phi_c) > U_\varepsilon(\phi_c, D_f)) < \delta$. This gives a contradiction, completing the proof.

We note that this theorem is true for any choice of candidate selection, as the proof assumes the candidate solution $\phi_c$ is arbitrary.

## F  Computing $U_\varepsilon(\phi, D_f)$ for Multi-class Sensitive Attributes

Suppose $S \in \mathcal{S}$ where $|\mathcal{S}| > 2$. To evaluate an estimate of $U_\varepsilon(\phi, D_f)$, we need to find the worst-case $\Delta_{\text{DP}}$.

We first feed $Z_i = \phi(X_i, S_i)$ for each data point in $D_f$ to $\tau_{\text{adv}}^*$. Since the adversarial predictor predicts the sensitive attribute, both the sensitive attribute and the label are multi-class. Thus, we obtain a predicted probability distribution of $\hat{Y}_i$ such that $\sum_{s \in \mathcal{S}} \Pr(\hat{Y}_i = s | Z_i) = 1$ (we apply *softmax* to the output of the multi-layer perceptron to get these probabilities). By splitting $D_f$ into $|\mathcal{S}|$ groups where according to their sensitive attributes, we can get an unbiased estimate (denoted as $\hat{p}^{(i)}(s|j)$) of $\Pr(\hat{Y} = s | S = j)$ with a sample $(X_i, S_i = j, \hat{Y}_i = s)$.

Following a similar procedure in Sec. 5.2.1, we draw $m$ unbiased estimates of $\Pr(\hat{Y} = s | S = j)$ for each $s, j \in \mathcal{S}$. Then we apply statistical tools (Student's t-test, for example) to construct a $1 - \delta/|\mathcal{S}|^2$ confidence interval (CI) $[c_l(s, j), c_u(s, j)]$ on $\Pr(\hat{Y} = s | S = j)$ with $\delta/(2|\mathcal{S}|^2)$ on both sides.

Finally, we set $U_\varepsilon(\phi, D_f) = \max_s(\max_j(c_u(s, j)) - \min_k(c_l(s, k))) - \varepsilon$.

We prove the correctness below.

**Theorem F.1.** *Suppose $\Pr(\hat{Y} = s | S = j)$ has $1 - \delta/|\mathcal{S}|^2$ confidence interval $[c_l(s, j), c_u(s, j)]$ with the confidence equally split on both sides for each $s, j \in |\mathcal{S}|$, and suppose $\hat{Y}$ is the optimal adversarial prediction that causes the maximum $\Delta_{DP}$, then*

$$\Pr[g_\varepsilon(\phi) \leq \max_s(\max_j(c_u(s, j)) - \min_k(c_l(s, k))) - \varepsilon] \geq 1 - \delta.$$

**Proof.** Suppose $\Pr(\hat{Y} = s | S = j)$ has $1 - \delta/|\mathcal{S}|^2$ confidence interval $[c_l(s, j), c_u(s, j)]$ with the confidence equally split on both sides, then $\Pr[\Pr(\hat{Y} = s | S = j) \leq c_l(s, j)] \leq \delta/(2|\mathcal{S}|^2)$ and $\Pr[\Pr(\hat{Y} = s | S = j) \geq c_u(s, j)] \leq \delta/(2|\mathcal{S}|^2)$ for each $s, j \in |\mathcal{S}|$.

By the union bound, we have $\Pr[\min_k \Pr(\hat{Y} = s | S = k) \leq \min_k c_l(s, k)] \leq \delta/(2|\mathcal{S}|)$ and $\Pr[\max_j \Pr(\hat{Y} = s | S = j) \geq \max_j c_u(s, j)] \leq \delta/(2|\mathcal{S}|)$ for each $s \in \mathcal{S}$.

By the union bound, we have $\Pr[\max_j \Pr(\hat{Y} = s | S = j) - \min_k \Pr(\hat{Y} = s | S = k) \geq \max_j c_u(s, j) - \min_k c_l(s, k)] \leq \delta/|\mathcal{S}|$ for each $s \in \mathcal{S}$.

By the union bound again, we have

$$\Pr[\max_s(\max_j \Pr(\hat{Y} = s | S = j) - \min_k \Pr(\hat{Y} = s | S = k)) \geq \max_s(\max_j c_u(s, j) - \min_k c_l(s, k))] \leq \delta.$$

Assuming the adversary is optimal, we have

$$g_\varepsilon(\phi) = \sup_\tau \Delta_{\text{DP}}(\tau, \phi) - \varepsilon \tag{39}$$

$$= \max_{j,k \in S} |\Pr(\hat{Y} = 1 | S = j) - \Pr(\hat{Y} = 1 | S = k)| - \varepsilon \tag{40}$$

$$= \max_{j \in S} \Pr(\hat{Y} = 1 | S = j) - \min_{k \in S} \Pr(\hat{Y} = 1 | S = k) - \varepsilon \tag{41}$$

$$\leq \max_{s \in S}(\max_{j \in S} \Pr(\hat{Y} = s | S = j) - \min_{k \in S} \Pr(\hat{Y} = s | S = k)) - \varepsilon, \tag{42}$$

where the predicted class $s$, the sensitive attribute classes $j$ and $k$ result in the worst-case differences over this group of samples.

| Dataset | Sensitive (# group) | Downstream tasks | Size | Pr(S) |
|---------|---------------------|------------------|------|-------|
| Adult | Gender (**2**) | Income, Gender | 41K | **0.332**, 0.668 |
| Health | Gender (**2**) | C.I., Age, Gender | 55K | **0.553**, 0.447 |
| Income | Marital Status (**5**) | Income, Marital Status | 195K | **0.52**, 0.02, 0.09, 0.02, 0.35 |

Table 1: Summary of dataset statistics. For each dataset, the first task is used for hyperparameter search and validation, and the last task is an adversarial task that predicts the sensitive attribute. The **bold** in the last column indicates the fraction of positive labels for the adversarial tasks. C.I. stands for Charlson Index.

Then

$$\Pr[g_\varepsilon(\phi) \geq \max_s(\max_j c_u(s,j) - \min_k c_l(s,k)) - \varepsilon] \tag{43}$$

$$\leq \Pr[\max_{s \in S}(\max_{j \in S} \Pr(\hat{Y} = s | S = j) - \min_{k \in S} \Pr(\hat{Y} = s | S = k)) - \varepsilon] \tag{44}$$

$$\geq \max_s(\max_j c_u(s,j) - \min_k c_l(s,k)) - \varepsilon] \tag{45}$$

$$\leq \delta. \tag{46}$$

Thus, $\Pr[g_\varepsilon(\phi) \leq \max_s(\max_j(c_u(s,j)) - \min_k(c_l(s,k))) - \varepsilon] \geq 1 - \delta$. $\square$

## G   Using Student's T-test to Construct Confidence Intervals

We construct the $1 - \delta$ confidence intervals of a random variable $p$ using Student's t-test provided $m$ samples $\hat{p}$ with the following steps: (1) Compute the sample mean $\bar{p} = \frac{1}{m} \sum_{k=1}^m \hat{p}^{(k)}$ where $k \in [1, \ldots, m]$; (2) Compute the sample standard deviation $\hat{\sigma} = \sqrt{\frac{1}{m-1} \sum_{k=1}^m (\hat{p}^{(k)} - \bar{p})^2}$; (3) Compute a $1 - \delta/2$ confidence lower bound $c_l$ and $1 - \delta/2$ confidence upper bound $c_u$ on $p_\varepsilon(\phi)$ using Student's t-test. That is $c_l = \bar{p} - \frac{\hat{\sigma}}{\sqrt{m}} t_{1-\delta/2, m-1}$ and $c_u = \bar{p} + \frac{\hat{\sigma}}{\sqrt{m}} t_{1-\delta/2, m-1}$ where $t_{1-\delta/2, m-1}$ is the $100(1 - \delta/2)$ percentile of the Student's t-distribution with $m - 1$ degrees of freedom. Student's t-test assumes the data to be normally distributed, and thus $m$ needs to be sufficiently large for the guarantees to hold (following the central limit theorem (CLT)).

## H   Different Techniques for Obtaining Confidence Bounds

The statistical confidence interval method in our framework is modular such that researchers can substitute alternatives depending on their domain-specific needs.

We note that prior work, including the fairness experiments in Thomas et al. [70] employed the Student's t-test for similar guarantees and demonstrated that the t-test yielded sufficiently conservative failure rates in practice. In our work, we observed similar behavior across datasets. Other than the student's t-test, we have explored Hoeffding's inequality [29], but found its bounds to be overly conservative for our datasets, limiting its practical utility.

Different statistical tools for obtaining confidence bounds can have different tradeoffs. For instance: one could consider bounds based on the Dvoretzky–Kiefer–Wolfowitz (DKW) inequality [3] which are less sensitive to distributional tails or empirical Bernstein bounds [49] which leverage sample variance. One could also explore bootstrap-based intervals, which empirically approximate sampling distributions.

We think that a rigorous comparison of confidence interval methods (e.g., parametric vs. non-parametric, bootstrap, etc.) is beyond the scope of this paper and merits its own study. Future work could systematically evaluate these alternatives to identify optimal bounds for specific applications.

## I   Datasets

The dataset statistics are listed in Table 1. The first dataset is the UCI *Adult* dataset [6],[2] which contains information of over 40,000 adults from the 1994 US Census. The sensitive attribute we consider is gender, and the targeted downstream task is to predict whether an individual earns more than $50K/year.

The second dataset is Heritage Health [34].[3] The targeted downstream task is to predict Charlson Comorbidity Index, and we consider gender as a sensitive attribute. We include an additional

---

[2] https://archive.ics.uci.edu/ml/datasets/Adult
[3] https://www.kaggle.com/c/hhp

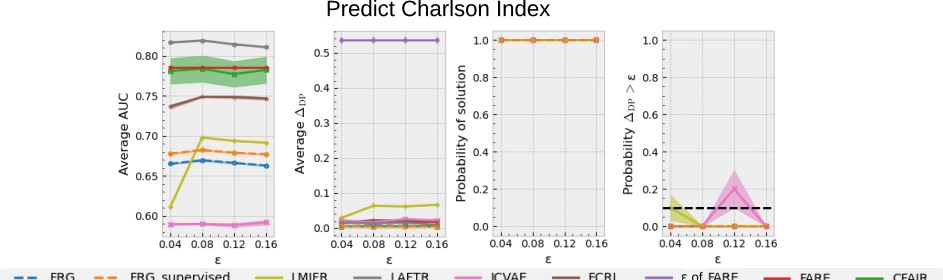

Figure 5: The evaluation on the Health dataset. The target label is Charlson Index. We vary $\varepsilon \in \{0.04, 0.08, 0.12, 0.16\}$. $\delta$ is fixed at 0.1.

downstream task that predicts age. Ages above 50 have positive labels, and ages below 50 have negative labels.

The third dataset we use is ACSIncome [17].[4] It includes data collected by the US Census across all states. We use the California dataset collected in 2018. The sensitive attribute is marital status, which has five classes: married, widowed, divorced, separated, and never married. The targeted downstream task is to predict whether an individual has income above \$50,000.

## J    Hyperparameter Tuning

In our hyperparameter tuning process, we adjust various parameters, including the step sizes (for the primary objective, the Lagrange multipliers, and the adversarial predictor), the initial Lagrange multipliers, the weight of the regularizers, the number of epochs, etc. The primary objective of hyperparameter tuning is not only to find a set of hyperparameters for the algorithm that minimizes $\Delta_{\text{DP}}$. Instead, our goal is to find hyperparameters that allow the algorithm to consistently provide a representation model that is $\varepsilon$-fair with as high expressiveness as possible. Thus, one should not tune hyperparameters separately for each of the training datasets we created. When we reuse the same training or validation set for hyperparameter search, we end up evaluating $\Delta_{\text{DP}}$ multiple times on the same training or validation set. As a result, $\Delta_{\text{DP}}$ evaluated on the model trained with the chosen hyperparameters may provide a biased estimation of $\Delta_{\text{DP}}$ on unseen future data. Consequently, the estimation of the probability $\Delta_{\text{DP}} \leq \varepsilon$ will also be biased. Therefore, we create additional datasets for hyperparameter tuning and adopt the same hyperparamters on different training datasets of the same size.

For baselines, we create validation sets by sampling 20% of the training data, while for FRG, we evaluate the models using the fairness test datasets (i.e., $D_f$ in Sec. 5.2). We tune each algorithm with grid search according to the metrics evaluated on the validation sets (for baselines) or on the fairness test sets (for FRG). For the Health dataset, as there are multiple downstream tasks, we only assume the Charlson Index labels are available for hyperparameter tuning.

For FRG and FRG_supervised, we set the hyperparameter $\alpha = 2$ in the main experiment. We provide a study of various $\alpha$'s in Appendix L.

Note that we set the minimum allowed step size for the primary objective to $10^{-6}$ and the minimum number of epochs to 100. This choice is motivated by the fact that an algorithm with an excessively small step size may have minimal impact on optimizing the primary objective and could potentially produce random representations that lack utility for downstream predictions, despite being highly likely to be fair.

We also note that we use the same number of dimensions for representation $Z$ ($Z = 50$ for Adult and Health and $Z = 100$ for Income) and the same hidden size for the downstream MLP for fair comparison. We use cross-entropy loss for all downstream models and Adam optimizer for all optimizations.

The detailed choices of hyperparameters for each of the datasets, the unfairness thresholds $\varepsilon$'s, and the baselines are provided with config files in the source code.

---

[4]`https://github.com/socialfoundations/folktables`

# K   The Tradeoff Between the Prediction Performance and Fairness

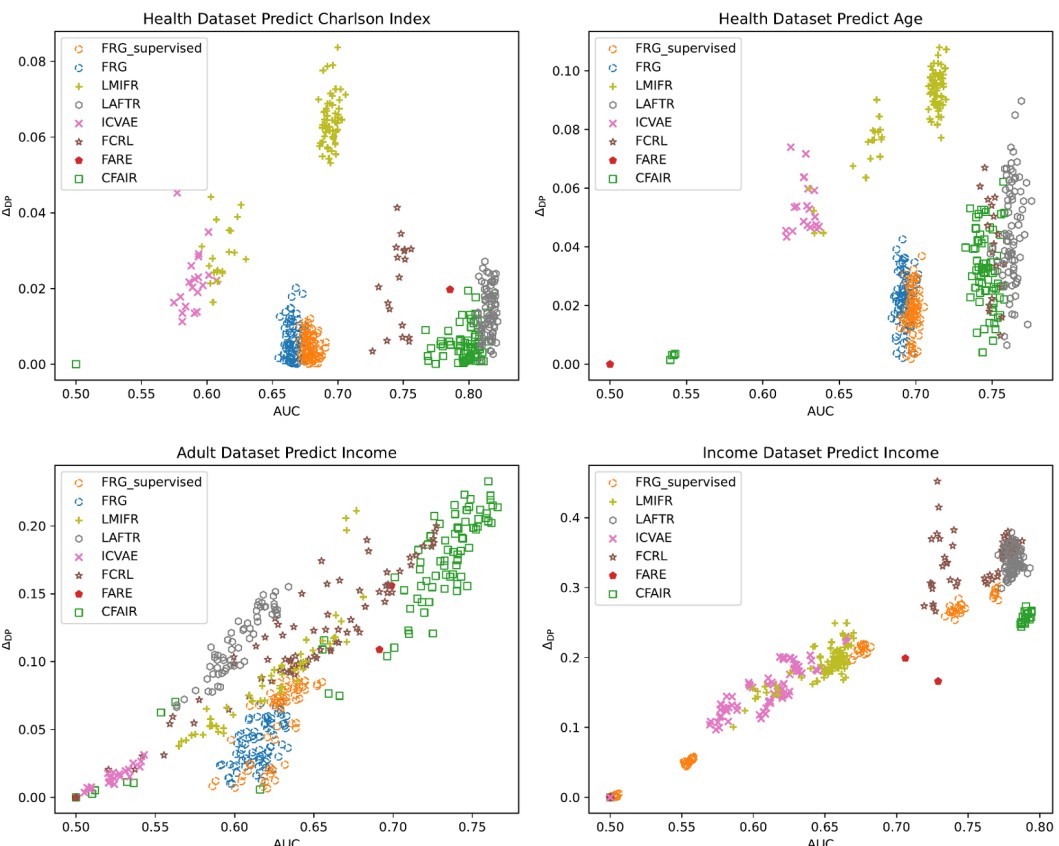

Figure 6: The tradeoff between AUC and $\Delta_{DP}$ on the Adult (bottom left), the Health (top left and top right) and the Income dataset (bottom right). FRG and FRG_supervised achieve the best tradeoff in Adult and are comparable to the best baselines in other datasets. Notice that even though some baselines achieve better tradeoff (e.g., FCRL, LMIFR, CFAIR in Health and CFAIR in income), they have high probabilities of violating the fairness constraints, especially in the adversarial downstream tasks (right Figures 3 and 4).

In Figure 6, the tradeoff between AUC and $\Delta_{DP}$ for each dataset is plotted. FRG and FRG_supervised achieve the best tradeoff in Adult and are comparable to the best baselines in other datasets. These results confirm that FRG and FRG_supervised are competitive in prediction performance while maintaining a low $\Delta_{DP}$. Other than FRG and FRG_supervised, the baselines that achieve optimal tradeoff for one downstream task could achieve worse $\Delta_{DP}$ in the adversarial tasks. For example, FCRL, CFAIR and LAFTR in the Health dataset and CFAIR in the Income dataset achieve the best tradeoff in targeted downstream tasks, but violate the fairness constraints consistently in the adversarial task (right Figures 3 and 4). There are only 1-3 points for FARE because their use of discrete distribution with finite support lowers the variability of the representations, which is an issue discussed in Comparison between FARE and FRG in Section 6.2.

## L   Evaluating the impact of $\alpha$

We highlight again that the choice of the confidence inflation hyperparameter $\alpha$ does not affect the validity of the high-confidence fairness guarantees. Here we evaluate different choices of $\alpha$'s in Figure 7 with $\delta = 0.1$. There are only minor differences in the performance and no impact on acceptance rates. Therefore, in this case, we may conclude that overfitting has not occurred. In our main experiment, we keep $\alpha = 2$ for all evaluations.

However, we think it is important to point out the potential overfitting if we do not inflate the confidence upper bound in candidate selection, especially when the constraint is restrictive. For example, when we set $\varepsilon = 0.035$ and $\delta = 0.01$, with results in Table 2, when $\alpha$ is closer to 1.0, it leads to a higher probability of returning NSF. This is the case when the candidate solution overfits

| $\alpha$ | Solution found | Avg. AUC | Avg. $\Delta_{DP}$ |
|---|---|---|---|
| 1.0 | 0.0 | - | - |
| 1.25 | 0.0 | - | - |
| 1.5 | 0.0 | - | - |
| 1.75 | 0.9 | 0.59 | 0.02 |
| 2.0 | 1.0 | 0.59 | 0.02 |
| 2.25 | 1.0 | 0.50 | 0.0 |
| 2.75 | 1.0 | 0.50 | 0.0 |
| 3.0 | 1.0 | 0.50 | 0.0 |

Table 2: On the Adult dataset with $\varepsilon = 0.035$ and $\delta = 0.01$.

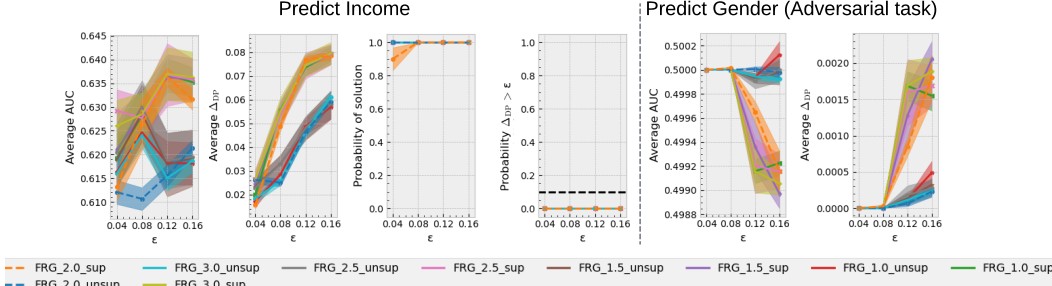

Figure 7: The study of the hyperparameter $\alpha$ on the Adult dataset with $\delta = 0.1$. We vary $\alpha \in 0.01, 0.05, 0.1, 0.15$. FRG_$\alpha$_sup and FRG_$\alpha$_unsup denotes FRG trained with and without supervision respectively.

the training data and overestimating the confidence that it can pass the fairness test. When $\alpha$ is larger it is more likely to return a solution. However, the performance can be affected because the candidate solution will be more conservative and may give a higher confidence than required to satisfy the fairness constraint.

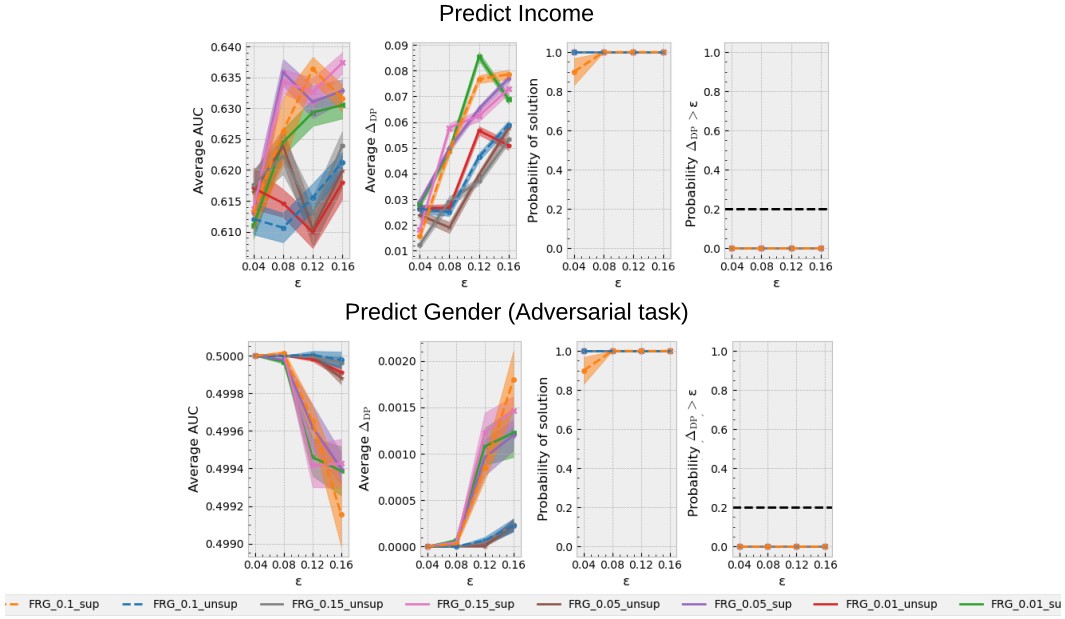

Figure 8: The study of the confidence level $\delta$ on the Adult dataset. We vary $\delta \in \{0.01, 0.05, 0.1, 0.15\}$. *FRG_$\delta$_sup* and *FRG_$\delta$_unsup* denotes FRG trained with and without supervision respectively.

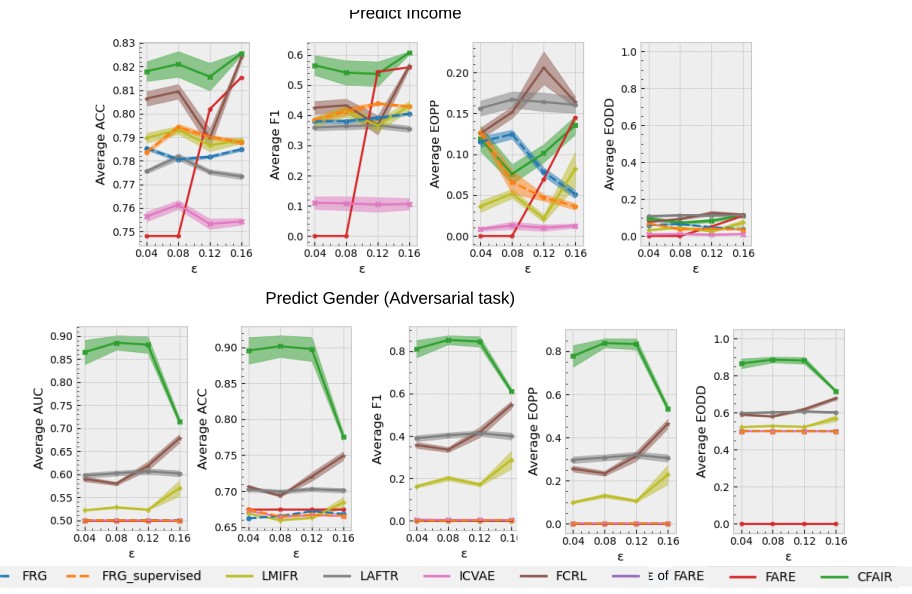

Figure 9: Additional metrics including F1, Average Accuracy (ACC), Equal Opportunity Difference (EOPP), Equalized Odds Difference (EODD) for the evaluation on the **Adult** dataset.

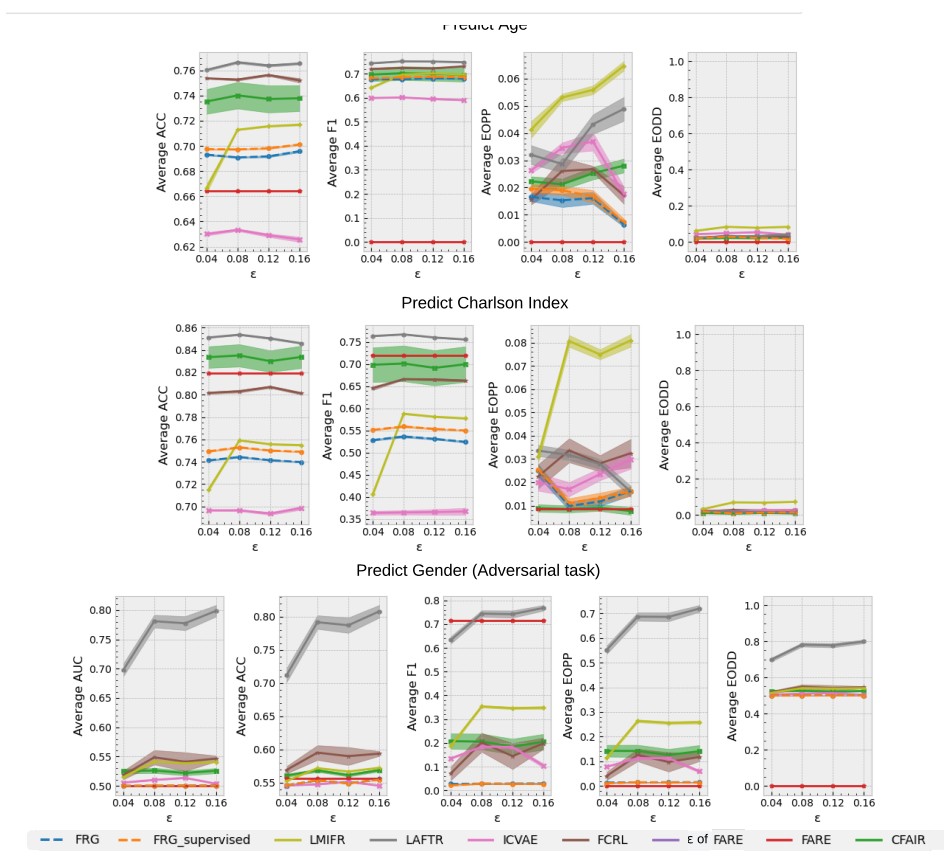

Figure 10: Additional metrics including F1, Average Accuracy (ACC), Equal Opportunity Difference (EOPP), Equalized Odds Difference (EODD) for the evaluation on the **Health** dataset.

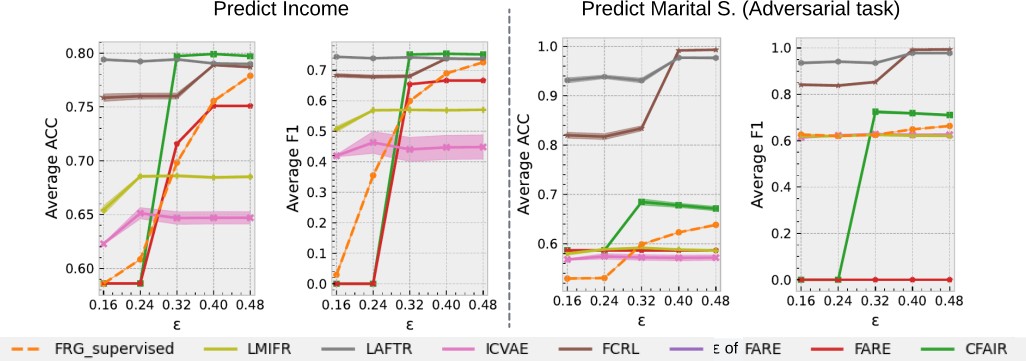

Figure 11: Additional metrics including F1, Average Accuracy (ACC) for the evaluation on the **Income** dataset.

# M   A Case Study: Using a Mutual Information-based Upper-bound for Constraining $\triangle_{\text{DP}}$ to Avoid the Assumption of Optimal Adversary

Different from the proposed method in the main text that uses the worst-case $\triangle_{\text{DP}}$ achieved by an oracle adversarial predictor to upper-bound $\triangle_{\text{DP}}$, we provide an alternative method that uses a mutual information (MI)-based upper bound for constraining $\triangle_{\text{DP}}$. Although the alternative method (named FRG-MI) does not rely on an oracle adversarial predictor for providing the guarantees, the upper-bound on $\triangle_{\text{DP}}$ is so loose that the method is shown impractical (the constraint is too conservative to provide good utility). In the following subsections, we will first introduce the MI-based bound on a strictly increasing convex function of $\triangle_{\text{DP}}$ as derived by [26] (Appendix M.1). As the alternative method shares similar components with FRG, including the candidate selection component and the fairness test component, we will document the changes to these components in Appendix M.2 and M.3 while referring to the main methods (Sec. 5) for the repeated details. In Appendix M.4, we prove that FRG-MI also provides a $1 - \delta$ confidence $\varepsilon$ fairness guarantees (Def. 4.2) as FRG does. In Appendix M.5, we evaluate FRG-MI empirically to show that it may not be suitable for practical use. Finally, we discuss several other alternatives one might consider for upper-bounding $\triangle_{\text{DP}}$ (Appendix M.6).

## M.1   Mutual information bounds demographic parity

The demographic-parity-based measure (Def. 3.1) is specified for downstream models. Since want our representation model to guarantee fairness for every possible downstream model and downstream task, we consider using a mutual information-based upper-bound on $\triangle_{\text{DP}}$. Gupta et al. [26] derived a bound for $\triangle_{\text{DP}}(\tau, \phi)$ that removes the dependency on the downstream model $\tau$. Specifically, Gupta et al. [26] showed that the mutual information between the representation and the sensitive attributes, denoted by $I(Z; S)$, can be used to limit the demographic parity of downstream models.

*Property* M.1 (Relation of mutual information to $\triangle_{\text{DP}}(\tau, \phi)$). For all downstream models $\tau$ in all downstream tasks,

$$I(Z; S) \geq \psi(\triangle_{\text{DP}}(\tau, \phi)),$$

where $\psi$ is a strictly increasing non-negative convex function derived by [26], and the details of which are in Appendix N.2. **Proof.** See the work of [26].

Notice that Property M.1 does not provide a direct upper bound on $\triangle_{\text{DP}}(\tau, \phi)$. Instead, it provides an upper bound on a strictly increasing non-negative convex function of $\triangle_{\text{DP}}(\tau, \phi)$. We use this property in the next section to guarantee fairness for representation models with high confidence.

## M.2   The modified fairness test compared to Section 5.2

Different from Section 5.2, we now avoid the adversarial predictor but use Property M.4 to develop a high-confidence upper-bound for $g_\varepsilon(\phi)$. In this section, we first develop $\tilde{g}_\varepsilon(\phi)$ where $\tilde{g}_\varepsilon(\phi) \leq 0$ only if $g_\varepsilon(\phi) \leq 0$, and propose evaluating $\tilde{g}_\varepsilon(\phi) \leq 0$ to provably determine whether a representation model $\phi$ is $\varepsilon$-fair, i.e., $g_\varepsilon(\phi) \leq 0$. We then follow similar procedure as Sec. 5.2 to construct a high-confidence upper bound on $\tilde{g}_\varepsilon(\phi)$ instead of $g_\varepsilon(\phi)$. We follow Sec. 5.2 for the evaluation process for a candidate solution $\phi_c$ using this high-confidence upper bound.

**A mutual-information-based evaluation.** Our goal is to evaluate whether $g_\varepsilon(\phi) \leq 0$ with high confidence. However, estimating $g_\varepsilon(\phi) = \sup_\tau \triangle_{\text{DP}}(\tau, \phi) - \varepsilon$ is intractable because it requires knowledge of all downstream models and all downstream tasks. To remove the dependency on downstream models, we apply Property M.1, and evaluate whether $I(Z; S) - \psi(\varepsilon) \leq 0$ instead of $\sup_\tau \triangle_{\text{DP}}(\tau, \phi) - \varepsilon \leq 0$ ($\psi$ is derived by Gupta et al. [26] and defined in Appendix N.2). Intuitively, when the mutual information between the representation and the sensitive attribute is low, it is hard for any model to predict $S$ given $Z$ with high accuracy. Therefore, any downstream model that does not explicitly aim to predict $S$ is even less likely to take advantage of the sensitive attribute to produce unfair predictions. Theoretically, evaluating $I(Z; S) - \psi(\varepsilon) \leq 0$ can provably determine the $\varepsilon$-fairness of $\phi$ under Def. 4.1. We postpone the theoretical analysis to Appendix. M.4.

Unfortunately, computing $I(Z; S)$ is intractable because it requires marginalizing the joint distribution of $(X, S, Z)$ over feature vector $X$, and so even this approach remains intractable. Multiple previous works have derived tractable upper bounds on $I(Z; S)$, which we discuss in detail in Appendix M.6.1. Let $\tilde{I}(Z; S)$ be one of these tractable upper bounds on $I(Z; S)$. Then, we define

$$\tilde{g}_\varepsilon(\phi) := \tilde{I}(Z; S) - \psi(\varepsilon). \tag{47}$$

With this upper bound, we now evaluate the $\varepsilon$-fairness of $\phi$ by evaluating $\tilde{g}_\varepsilon(\phi) \leq 0$. In Lemma M.2, we prove if $\Pr\left(\tilde{g}_\varepsilon(a(D)) \leq 0\right) \geq 1 - \delta$, then algorithm $a$ provides the desired high-confidence fairness guarantee.

$1 - \delta$ **confidence upper bound on $\tilde{g}_\varepsilon(\phi)$.** We follow two steps similar to Sec. 5.2 to compute a $1 - \delta$ confidence upper bound on $\tilde{g}_\varepsilon(\phi)$. (1) Obtain $m$ i.i.d. unbiased estimates $\hat{g}^{(1)}, \ldots, \hat{g}^{(m)}$ of $\tilde{g}_\varepsilon(\phi)$ using $D_f$, i.e., $\mathbb{E}[\hat{g}^{(j)}] = \tilde{g}_\varepsilon(\phi)$ for any $j \in [1, ..., m]$. (2) Apply standard statistical tools such as Student's t-test [69] or Hoeffding's inequality [29] to construct a $1 - \delta$ confidence upper bound on $\tilde{g}_\varepsilon(\phi)$ using $\hat{g}^{(1)}, \ldots, \hat{g}^{(m)}$. We also use Student's t-test for our experiments (Appendix M.5).

Similar to Sec. 5.2, we define $U'_\varepsilon : (\Phi, \mathcal{D}) \to \mathbb{R}$ to be such a function that produces a $1 - \delta$ confidence upper bound. Specifically, for $U'_\varepsilon(\phi, D_f)$, we have the following,

$$\Pr\left(\tilde{g}_\varepsilon(\phi) \leq U'_\varepsilon(\phi, D_f)\right) \geq 1 - \delta. \tag{48}$$

The remaining steps for evaluating the candidate solution are equivalent to those of Sec. 5.2.

### M.3 The modified candidate selection compared to Section 5.3

The candidate selection procedure is not changed from Section 5.3 except now it estimates the alternative high-confidence upper bound $U'_\varepsilon$ (Def.48). We can also avoid the adversarial training process for estimating the upper bound as we can adopt the same procedure as in the fairness test.

### M.4 Theoretical analysis

In this section we prove that FRG-MI is a representation learning algorithm that provides the desired high confidence $\varepsilon$-fairness guarantee, i.e., the probability that it produces a representation that is not $\varepsilon$-fair for every downstream task and model is at most $\delta$.

We first prove in Lemma M.2 that if an algorithm $a$ satisfies $\Pr\left(\tilde{g}_\varepsilon(a(D)) \leq 0\right) \geq 1 - \delta$, then algorithm $a$ provides the $1 - \delta$ confidence $\varepsilon$-fairness guarantee described in Def. 4.2. We then prove in Theorem M.3 that FRG-MI indeed satisfies $\Pr\left(\tilde{g}_\varepsilon(a(D)) \leq 0\right) \geq 1 - \delta$. Altogether, we can conclude that FRG guarantees with $1 - \delta$ confidence that $\Delta_{\mathrm{DP}}(\tau, a(D))$ is upper-bounded by $\varepsilon$ for any $\tau$ (recall that here $a$ corresponds to FRG-MI and $a(D)$ corresponds to the representation model parameters returned by FRG-MI when run on dataset $D$).

**Lemma M.2.** *If algorithm $a$ satisfies $\Pr\left(\tilde{g}_\varepsilon(a(D)) \leq 0\right) \geq 1 - \delta$, then algorithm $a$ provides the $1 - \delta$ confidence $\varepsilon$-fairness guarantee described in Def. 4.2.*

**Proof.** *Suppose $\Pr\left(\tilde{g}_\varepsilon(a(D)) \leq 0\right) \geq 1 - \delta$. By Eq. 47, $\tilde{g}_\varepsilon(a(D)) = \tilde{I}(Z; S) - \psi(\varepsilon) \geq I(Z; S) - \psi(\varepsilon)$. By property M.1, $I(Z; S) \geq \sup_\tau \psi(\Delta_{DP}(\tau, a(D)))$. So, the event $(\tilde{g}_\varepsilon(a(D)) \leq 0)$ implies that$(I(Z; S) - \psi(\varepsilon) \leq 0)$, which further implies $(\sup_\tau \psi(\Delta_{DP}(\tau, a(D))) - \psi(\varepsilon) \leq 0)$. Using the fact that $\psi$ is strictly increasing in $[0, 1]$ (Appendix N.2), we have the following equivalent events:*

$$\left(\sup_\tau \psi(\Delta_{DP}(\tau, a(D))) - \psi(\varepsilon) \leq 0\right) \tag{49}$$

$$\Longleftrightarrow \left(\psi(\sup_\tau \Delta_{DP}(\tau, a(D))) \leq \psi(\varepsilon)\right) \tag{50}$$

$$\Longleftrightarrow \left(\sup_\tau \Delta_{DP}(\tau, a(D)) \leq \varepsilon\right) \tag{51}$$

$$\Longleftrightarrow \left(\sup_\tau \Delta_{DP}(\tau, a(D)) - \varepsilon \leq 0\right) \tag{52}$$

$$\Longleftrightarrow \left(g_\varepsilon(a(D)) \leq 0\right). \tag{53}$$

*It follows that $\Pr\left(g_\varepsilon(a(D)) \leq 0\right) \geq \Pr\left(\tilde{g}_\varepsilon(a(D)) \leq 0\right) \geq 1 - \delta$. So, FRG-MI (algorithm $a$) provides the desired $1 - \delta$ confidence $\varepsilon$-fairness guarantee described in Def. 4.2, completing the proof.*

**Theorem M.3.** *FRG-MI provides the $1 - \delta$ confidence $\varepsilon$-fairness guarantee described in Def. 4.2.*

**Proof.** By Lemma M.2, if FRG-MI satisfies $\Pr\left(\tilde{g}_\varepsilon(a(D)) \leq 0\right) \geq 1 - \delta$, then FRG-MI provides the desired $1 - \delta$ confidence $\varepsilon$-fairness guarantee. Following the same proof for Theorem 5.3 in Appendix E, we can prove by contradiction that when $a$ represents FRG-MI, $\Pr\left(\tilde{g}_\varepsilon(a(D)) \leq 0\right) \geq 1 - \delta$.

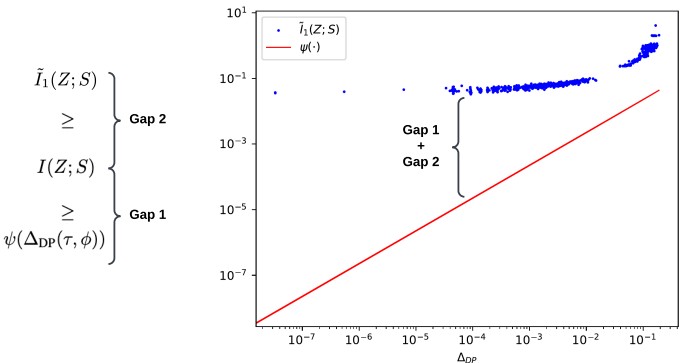

Figure 12: Using the *Adult* dataset (details in Appendix. I), we run L-MIFR [68] with different hyper-parameters to find representation models that achieve different $\Delta_{\mathrm{DP}}(\tau, \phi)$. For each of the representation model, we record the corresponding tractable upper bound to $I(Z; S)$ by Song et al. [68, Section 2.2], denoted as $\tilde{I}_1(Z; S)$, and make the scatter plot in blue. We plot the function $\psi(\cdot)$ (Appendix N.2) in red. We highlight that there exists a gap between $\tilde{I}_1(Z; S)$ and $\psi(\Delta_{\mathrm{DP}}(\tau, \phi))$, which consists of two gaps, $\tilde{I}_1(Z; S) - I(Z; S)$ and $I(Z; S) - \psi(\Delta_{\mathrm{DP}}(\tau, \phi))$, and it can be observed empirically as shown by the plot. As $\Delta_{\mathrm{DP}}$ decreases, the gap between $\tilde{I}_1(Z; S)$ and $\psi(\Delta_{\mathrm{DP}}(\tau, \phi))$ increases.

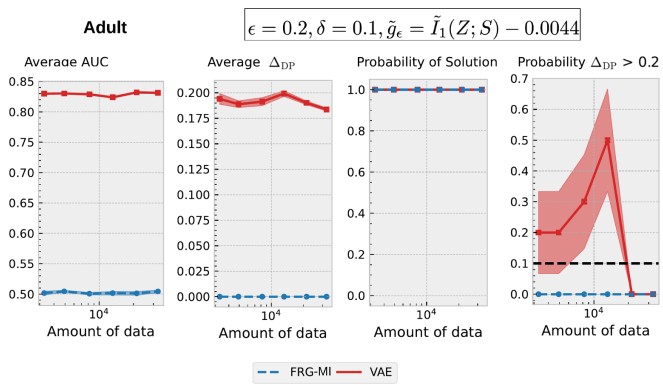

Figure 13: We give an example of employing FRG-MI to provide high-confidence fairness guarantees (Def. 4.2) on the *Adult* dataset, including VAE as a baseline.

## M.5   Experiments

We first visualize in Fig. 12 and confirm that the gap between $\tilde{I}_1(Z; S)$ and $\psi(\Delta_{\mathrm{DP}}(\tau, \phi))$ indeed exists empirically and the gap increases as $\Delta_{\mathrm{DP}}$ approaches 0.

We then evaluate FRG-MI that provides high-confidence fairness guarantees (Def. 4.2) on the *Adult* dataset. For demonstration purposes, we select $\varepsilon = 0.2$ and $\delta = 0.1$, which means that FRG-MI guarantees with 90% confidence that downstream models do not violate $\Delta_{\mathrm{DP}}(\tau, \phi) \leq 0.2$. It is worth noting that the selected $\varepsilon = 0.2$ is smaller than both the $\Delta_{\mathrm{DP}}$ calculated with the true labels (0.26), and the upper bound on $\Delta_{\mathrm{DP}}$ calculated with the prediction labels from a predictor that achieves equalized odds [76, Theorem 3.1]. We estimate $\Pr(S = 1) \approx 0.668$ from the dataset, which yields $\psi(\varepsilon) \approx 0.0044$. We incorporate the constraint $\tilde{I}_1(Z; S) \leq \psi(\varepsilon)$ to guarantee $\varepsilon$-fairness with $1 - \delta$ confidence ($\tilde{I}_1(Z; S)$ denotes the upper bound to $I(Z; S)$ as derived by Song et al. [68, Section 2.2]). We include a vanilla VAE without any fairness consideration as a baseline. The amount of training data used varies from 10%, 15%, 25%, 40%, 65% to 100% of the original data.

We show the result in Fig. 13. As demonstrated in the second and fourth plots, FRG-MI violates the constraint $\Delta_{\mathrm{DP}}(\tau, \phi) \leq 0.2$ with a probability smaller than 0.1, whereas VAE violates the constraint with a probability larger than 0.1 when it uses less than 65% of the training data. According to the third plot, FRG-MI can also return solutions (i.e., not NSF) for all the trials.

Nonetheless, the constraint $\tilde{I}_1(Z; S) \leq \psi(\varepsilon)$ is overly conservative, which leads to relatively low AUC on average, as illustrated in the first plot. Additionally, the fourth plot demonstrates FRG-MI's ability to consistently keep $\Delta_{\text{DP}}$ near zero. Hence, applying an even stricter $\varepsilon$ constraint on FRG-MI for high-confidence fairness guarantees is impractical and unnecessary.

We further analyze the gap between $I(Z; S)$ and $\psi(\sup_\tau \Delta_{\text{DP}}(\tau, \phi))$ in Appendix O, and the gap between $\tilde{I}_1(Z; S)$ and $I(Z; S)$ in Appendix P.

### M.6 Other alternatives for upper-bounding $\Delta_{\text{DP}}$

So far we have discussed using mutual information to upper bound $\Delta_{\text{DP}}$ (the violation of the demographic parity constraint), and ensure the $\varepsilon$-fairness of a representation model with high confidence (Sec. 5.2). Since $I(Z; S)$ is intractable, in Appendix M.6.1 we review four tractable upper bounds on $I(Z; S)$, and also discuss why in our experiments we adopt the first upper bound, $\tilde{I}_1(Z; S)$, derived by Song et al. [68, Section 2.2]. We then test whether $\tilde{I}_1(Z; S) \leq \psi(\varepsilon)$ to obtain the desired fairness guarantee (Eq. 47).

Because mutual information can be intractable, one might consider alternative methods for bounding $\Delta_{\text{DP}}$. In Appendix M.6.2, we explore potential alternatives for upper bounding $\Delta_{\text{DP}}$ but find limitations that prevent the adoption of these methods in FRG.

### M.6.1 The Tractable Upper Bounds on $I(Z; S)$

To our best knowledge, there are four tractable upper bounds on mutual information $I(Z; S)$ as derived by previous work [26, 54, 68]. Next, we discuss these approaches and their limitations. Although our general approach is compatible with any upper bound on mutual information, given the limitations of each method, we consider the first of the two approaches ($\tilde{I}_1(Z; S)$ below) by Song et al. [68] the most suitable in practice. Thus, we only adopt $\tilde{I}_1(Z; S)$ in our experiments.

Song et al. [68] proposed two upper bounds on $I(Z; S)$.

$\tilde{I}_1(Z; S)$: **the first upper bound derived by [68, Section 2.2].** We denote the first upper bound as $\tilde{I}_1(Z; S)$ and $\tilde{I}_1(Z; S) \geq I(Z; X, S) \geq I(Z; S)$ [68, Section 2.2]. This is a theoretically guaranteed upper bound. We discuss the limitation of this upper bound in Appendix P that using this upper bound may diminish the expressiveness of the representations. However, we still find it effective for FRG to limit $\Delta_{\text{DP}}$ by $\varepsilon$ in experiment (Sec. 6).

$\tilde{I}_2(Z; S)$: **the second upper bound derived by [68, Section 2.3].** Song et al. [68] proposed a tighter upper bound compared to $\tilde{I}_1(Z; S)$, which we denote as $\tilde{I}_2(Z; S)$. However, it requires adversarial training, and the true upper bound can only be obtained when the adversarial model approaches global optimality. This is not ideal because if the adversarial model is under-performing, we may under-estimate the upper bound to $I(Z; S)$, and guaranteeing $\tilde{I}_2(Z; S) \leq \psi(\varepsilon)$ does not guarantee $I(Z; S) \leq \psi(\varepsilon)$ or $\varepsilon$-fairness. This result has also been confirmed by prior work including Elazar and Goldberg [21], Gupta et al. [26], Xu et al. [73] and Gitiaux and Rangwala [23].

$\tilde{I}_3(Z; S)$: **the upper bound derived by Moyer et al. [54].** Moyer et al. [54] found $I(Z; S) = I(Z; X) - H(X|S) + H(X|Z, S)$ where $H$ denotes entropy. They proposed ignoring the unknown positive constant term $H(X|S)$ and using the reconstruction error, i.e., $-\mathbb{E}_{q_\phi(Z|X,S)}\left[\log p_\theta(X|Z, S)\right]$ to be an upper bound of $H(X|Z, S)$ [54, Equations 2–7]. Let $\tilde{I}_3(Z; S) := I(Z; X) - \mathbb{E}_{q_\phi(Z|X,S)}\left[\log p_\theta(X|Z, S)\right]$. Moreover, it can be difficult to estimate the gap $\tilde{I}_3(Z; S) - I(Z; S)$ because (1) $H(X|S)$ is hard to estimate; (2) $\tilde{I}_3(Z; S)$ is sensitive to the performance of the reconstruction model.

$\tilde{I}_4(Z; S)$: **the upper bound derived by Gupta et al. [26].** Gupta et al. [26] observed that $I(Z; S) = I(Z; S|X) + I(Z; X) - I(Z; X|S)$. They then derived a lower bound for the term $I(Z; X|S)$ using constrative estimation so that $I(Z; S)$ can be upper-bounded. Specifically, they proved $I(Z; X|S) \geq \mathbb{E}_{p(X,Z,S)}\left[\log \frac{e^{f(X,Z,S)}}{\frac{1}{M}\sum_{m=1}^{M} e^{f(X_m, Z, S)}}\right]$, where $p(X, Z, S)$ is the joint distribution of $(X, Z, S)$, $X_1, \cdots, X_M \sim p_{X|S}$, $p_{X|S}$ is the conditional distribution of $X$ given $S$, and $f$ is an arbitrary function [26, Proposition 5]. Since the distribution $p_{X|S}$ is unknown, the authors use the $X, S$ pairs in the dataset as samples from this conditional distribution. When making a point estimate of the expectation, they use one sample from the dataset to evaluate the numerator, and use $M$ samples from the same dataset to evaluate the denominator. This means that the estimation of

the expectation can be biased because the point estimates are not independent. Empirically, we also observe this issue and find that it tends to result in over-estimates of $I(Z; X|S)$ and under-estimates of $I(Z; S)$. Given how these terms are used in the expression for $I(Z; S)$, this results in bounds on mutual information that do not hold.

### M.6.2 Alternative Methods for Upper-bounding $\Delta_{\text{DP}}$

One might consider alternative methods for bounding $\Delta_{\text{DP}}$ because mutual information can be intractable and there can be a significant gap between mutual information and $\psi(\Delta_{\text{DP}})$ (that is, the upper bound can be loose). Several alternative methods have been proposed, which can provide bounds on $\Delta_{\text{DP}}$ using bounds on the total variation between the conditional distributions $p_{\tau,\phi}(\hat{Y}|S = 0)$ and $p_{\tau,\phi}(\hat{Y}|S = 1)$ [5, 48, 66, 76]. However, to our knowledge, there is not a known function such as $\psi$ (Appendix N.2) that expresses the relation between the total variation and demographic parity, so total variation cannot be used to upper bound $\sup_\tau \Delta_{\text{DP}}(\tau, \phi)$ with a specific $\varepsilon$. In other work, Jovanović et al. [33, Section 5] proposed a practical certificate that upper bounds $\sup_\tau \Delta_{\text{DP}}(\tau, \phi)$. However, their method requires $Z$ to be a discrete random variable, which is restrictive for general representation learning. Therefore, these methods are not suitable for our framework as they cannot be used to learn $\varepsilon$-fair representation models with a high-confidence guarantee.

## N    Details of Property M.1

Gupta et al. [26] has derived Property M.1 where $I(Z; S)$ is an upperbound for a strictly increasing non-negative convex function in $\Delta_{\text{DP}}$ of any $\tau$, which we denote as $\psi$. Gupta et al. [26] has also found that when $I(Z; S) = 0$, $\psi(\Delta_{\text{DP}}(\tau, \phi)) = 0$ and $\Delta_{\text{DP}}(\tau, \phi) = 0$. We now define $\psi$ in detail by first introducing a helper function $f$.

**Definition N.1** (A helper function $f$).

$$f(V) = \max\left(\log\left(\frac{2 + V}{2 - V}\right) - \frac{2V}{2 + V}, \frac{V^2}{2} + \frac{V^4}{36} + \frac{V^6}{288}\right).$$

with domain $V \in [0, 2)$.

**Definition N.2** (function $\psi$ with parameter $\Delta_{\text{DP}}(\tau, \phi)$). When $S$ is binary, and $f$ follows Def. N.1,

$$\psi(\Delta_{\text{DP}}(\tau, \phi)) = (1 - \pi)f(\pi\Delta_{\text{DP}}(\tau, \phi)) + \pi f((1 - \pi)\Delta_{\text{DP}}(\tau, \phi))$$

where $\pi = P_s(S = 1)$ with $P_s$ as the marginal distribution of $S \in \{0, 1\}$.

When $S$ is multinomial with $K$ classes,

$$\psi(\Delta_{\text{DP}}(\tau, \phi)) = f(\alpha\Delta_{\text{DP}}(\tau, \phi)), \alpha = \min_{k=1,\dots,K} \pi_k,$$

where $\pi_k = P_s(S = k)$ with $P_s$ as the marginal distribution of $S \in \{1, \dots, K\}$.

## O    The Non-trivial Gap between $I(Z; S)$ and $\psi(\sup_\tau \Delta_{\text{DP}}(\tau, \phi))$

In this section, we analyze the non-trivial gap between $I(Z; S)$ and $\psi(\sup_\tau \Delta_{\text{DP}}(\tau, \phi))$ that makes it difficult for any algorithm to obtain $\varepsilon$-fairness.

As shown by Gupta et al. [26, Figure 6], there tends to be a significant gap between $I(Z; S)$ and $\psi(\sup_\tau \Delta_{\text{DP}}(\tau, \phi))$. Using their Figure 6 as an example, when $I(Z; S) \approx 0.035$, $\Delta_{\text{DP}}(\tau, \phi) \approx 0.15$ and $\psi(\Delta_{\text{DP}}(\tau, \phi)) \approx 0.0025$. So, to ensure that $\Delta_{\text{DP}}(\tau, \phi) \leq 0.15$ with high confidence using the $\psi$-based bound on mutual information, one must ensure that $I(Z; S) \leq 0.0025$ with high confidence. However, in reality ensuring that $\Delta_{\text{DP}}(\tau, \phi) \leq 0.15$ only requires $I(Z; S) \leq 0.035$. Obtaining a solution that satisfies $I(Z; S) \leq 0.0025$ is far more difficult than obtaining a solution that satisfies $I(Z; S) \leq 0.035$, and hence using the $\psi$-based bound on mutual information results in exceedingly conservative bounds on $\Delta_{\text{DP}}$.

## P    The Non-trivial Gap between $\tilde{I}_1(Z; S)$ and $I(Z; S)$

In this section we analyze the non-trivial gap between $\tilde{I}_1(Z; S)$ and $I(Z; S)$ where $\tilde{I}_1(Z; S)$ (Appendix M.6.1) is one of the upper bounds to $I(Z; S)$ as derived by Song et al. [68, Section 2.2]. We begin by analyzing the gap between $I(Z; X, S)$ and $I(Z; S)$. $I(Z; X, S) - I(Z; S) =$

$H(Z|S) - H(Z|X, S) = H(X|S) - H(X|Z, S) = I(X; Z|S)$. This is the mutual information between $X$ and $Z$ given $S$, which is closely related to the primary objective we hope to maximize. Overall, we have the following:

$$I(Z; S) \leq I(Z; X, S) \tag{54}$$

$$= I(Z; S) + I(X; Z|S) \tag{55}$$

$$\leq \tilde{I}_1(Z; S) \tag{56}$$

Although using a constraint $\tilde{I}_1(Z; S) \leq \psi(\varepsilon)$ encourages both $I(Z; S)$ and $I(X; Z|S)$ to be small which seems to diminish the expressiveness of the representation model, we show empirically that it is effective for upper bounding mutual information and the $\Delta_{\text{DP}}$ of the downstream tasks with high probability in experiment (Sec. M.5).

