# OpenReview forum: "Fair Representation Learning with Controllable High Confidence Guarantees via Adversarial Inference"
_NeurIPS.cc/2025/Conference — NeurIPS 2025 poster_

### Official Review · Reviewer_5EpJ · 2025-06-30

**Clarity:** 3
**Significance:** 3
**Originality:** 3
**Rating:** 4
**Confidence:** 1

**Summary:**

The paper proposes FRG, a framework for fair representational learning that guarantee specific high confidence fairness bounds for any downstream task. The works achieves this by proposing a adversarial inference component and statistic fairness test.

**Questions:**

Please see the question in the weakness field (included for clarity).

**Ethical Concerns:**

["NO or VERY MINOR ethics concerns only"]

**Final Justification:**

The authors have responded to my queries. I am not an expert in this field and therefore I will retain my assessments.

**Limitations:**

Yes.

**Quality:**

3

**Strengths And Weaknesses:**

## Strength

1. The authors addresses a gap in the research field, indicating that existing methods provide empirical or estimates fairness guarantees without formalized proof.

2. The authors' theoretical proof between demographic parity and covariance seems sounds, with reasonable foundation for adversarial inference

3. The author provides sufficiently strong empirical evaluation

## Weakness

Although I am not a expert in this field:

1. The authors strong emphasis on demographic parity seems limiting, noting that there exist other popular fairness measurements such as equal opportunity and equalized odds (multiple fairness measurements are often considered in the fairness measure literature for such works). How does this work fair for other fairness measurements?

2. The authors proposes a strong IID assumption of the data. Although the assumption is fair for theoretical study, I am concern on the distribution shift between training and deployment. More empirical test could be provided to address this concern.

3. The authors proposed solution requires a multi-stage pipeline, which could potential be very computational heavy for large and datasets and complex task. How scalable is such a solution?

4. Additionally, how should one consider the thresholds $\epsilon$ and $\delta$?

---

> ### Author Rebuttal · Authors · 2025-07-30
>
> Thank you so much for carefully reviewing our work and appreciating our novelty, the empirical and theoretical results. We address your concern as follows.
>
> ---
>
> ## Q1: How can this work be extended for other fairness measurements including equal opportunity and equalized odds?
> This is an insightful question. Indeed, the proposed method can be ***applied to Equal Opportunity (EOP) and equalized odds (EOD) with simple modification***, but it requires that the outcome labels, Y, for the downstream task of concern are available to compute EOP and/or EOD. We will use EOP as an example. $\Delta_{\text{EOP}}(\tau,\phi) \coloneqq |\Pr(\hat{Y} = 1 |S =1,Y=1) -  \Pr(\hat{Y} = 1 | S = 0,Y=1)|$. This is equivalent to $\Delta_{\text{DP}}(\tau,\phi|Y=1)$. The rest follows, except we only consider the subset of the data where $Y=1$ when constructing the high-confidence upper-bound. Similarly, $\Delta_{\text{EOD}}(\tau,\phi) = \max(\Delta_{\text{DP}}(\tau,\phi|Y=0), \Delta_{\text{DP}}(\tau,\phi|Y=1))$ which one can also construct the high-confidence upper-bound using our framework by splitting $\delta$ in half for each of $\Delta_{\text{DP}}(\tau,\phi|Y=0)$ and $\Delta_{\text{DP}}(\tau,\phi|Y=1).$
>
> In the revised manuscript, we will clarify how our approach can be used to constrain EOP and EOD. However, we note that the high-confidence upper bound requires outcome labels, Y, from a specific downstream task, so in this context the guarantee can be applied only to that task, which might not be generalized to arbitrary downstream tasks, as it is the case for the DP (demographic parity). This is a common limitation to existing approaches as well (e.g., [1-2]).
>
> #### [1] Jovanović et al. Fare: Provably fair representation learning with practical certificates. ICML 23.
> #### [2] Song et al. Learning controllable fair representations. AISTATS 19.
>
> ---
>
> ## Q2: The IID assumption of the data is a concern to the distribution shift between training and deployment.
> We think this is an insightful question. We assume the data are i.i.d. and we agree it could be a concern to some distribution shift. If the test dataset has a different distribution of $(X,S)$, the guarantee that the representation model is fair with high probability may no longer hold. However, the guarantee is robust to label shifts because the label is not necessary when learning the representation model and constructing the high-confidence upper bounds. If the test dataset has a different distribution of label $Y$ or distribution of $Y$ conditioned on $(X,S)$, the guarantee will still hold.
>
> We think that addressing different kinds of distribution shifts of input $(X,S)$ is beyond the scope of our work. Future work can study approaches to construct a high-confidence upper bound with the provided knowledge of the distribution shifts. For example, the shift could be bounded, the shift may occur only to $S$ but not to $X$, the shift may be periodic, etc.
>
> We will include this discussion in our revision.
>
> ---
>
> ## Q3: The authors proposed solution requires a multi-stage pipeline, which could potential be very computational heavy for large and datasets and complex task. How scalable is such a solution?
> Our method requires additional steps for learning the adversary and for performing the fairness test compared to traditional methods. While the training of the adversary and the fairness test introduce some computational overhead, they are ***very small*** (around 5% of the overall runtime) compared to the training time of the representation model (95% of the runtime). Here is an example for the Income dataset. We train the representation models with 500 epochs across all baselines. The overall runtime of FRG is 330s, which includes training the adversary, representation model, and the fairness test. Training the adversarial model takes 16s (4.8%), the fairness test takes only 0.06s (0.02%), while training the representation takes about 95% of the runtime. Here is a comparison with the baselines: CFAIR takes 303s, ICVAE takes 1021s, LMIFR takes 336s, LAFTR takes 326s, FARE takes 41s (only one call to fit the decision tree). Thus, we think the computation cost is not a concern. In the revised manuscript, we will include a short paragraph clarifying this.
>
> ---
>
> ## Q4: How should one consider the thresholds $\epsilon$ and $\delta$?
> This is an insightful question. Intuitively, our framework guarantees that any (including adversarial) downstream tasks and models using the representations generated by its learned representation model will have $\Delta_{\text{DP}} <= \epsilon$ with probability at least 1-\delta. How to choose thresholds $\epsilon$ and $\delta$ will vary case by case. If one wants the guarantee to hold with a high probability, one should set a small $\delta$. If one wants to enforce a strict parity constraint, one should set a small $\epsilon$.
>
> ---
>
> Thank you again for reviewing the work, and we will be happy to discuss any further questions you may have.

---

> > ### Author Response · Authors · 2025-08-05
> > **Discussion regarding the rebuttal**
> >
> > Dear Reviewer 5EpJ,
> >
> > We are writing to follow up on our rebuttal. As we are approaching the end of the discussion period, we would like to check if our response has sufficiently addressed the concerns you raised. We would be grateful for any feedback and are happy to answer any further questions.
> >
> > Thank you so much for your help in improving our work.

---

> > > ### Comment · Reviewer_5EpJ · 2025-08-06
> > > **Rebuttal Response**
> > >
> > > Thank you for the detailed response to my question. The authors have address all of my question for now.

---

> > > > ### Author Response · Authors · 2025-08-08
> > > > **Thank you!**
> > > >
> > > > Thank you for taking the time to read and respond to our rebuttal and for your positive feedback. We will be sure to incorporate the discussions and clarifications above in the final version.

---

### Official Review · Reviewer_XhGo · 2025-07-03

**Clarity:** 2
**Significance:** 3
**Originality:** 3
**Rating:** 4
**Confidence:** 3

**Summary:**

The paper introduces Fair Representation learning with high confidence Guarantees(FRG), a framework that learns fair representations with high confidence. The fairness tolerance and confidence level are user-specified. The framework provides this guarantee for any downstream task and model that uses the learned representation. FRG splits the training dataset into a candidate-selection set and a fairness-test set. Then, FRG uses the candidate selection set to train a representation learning model. The paper utilizes the VAE-style objective and adds a fairness proxy constraint to ensure the learned representations satisfy the fairness constraint. After the candidate model is provided, FRG trains an adversarial model that tries to estimate sensitive attributes using the learned representation. The authors show that the adversary that maximizes the covariance between prediction and sensitive attribute is equivalent to maximizing the demographic parity(DP) gap. Lastly, the framework uses the fairness-test set to provide confidence bounds on the worst-case DP gap using the Student t-test intervals. If the test fails, the framework returns that no solution was found(NSF). The experiments cover multiple real-world datasets, multiple tasks, and comparisons with baselines.

**Questions:**

Please see the weaknesses section.

**Ethical Concerns:**

["NO or VERY MINOR ethics concerns only"]

**Final Justification:**

I think that this paper has more positive sides than negative sides. The authors replied to all reviewers in detail.

**Limitations:**

yes

**Paper Formatting Concerns:**

No major formatting issues.

**Quality:**

3

**Strengths And Weaknesses:**

Strengths
- To the best of my knowledge, this is the first fair representation learning method offering user-controllable fairness tolerance and confidence certification on any downstream task that is required to be DP fair.
- The experiments section is quite comprehensive, the authors compare their method with multiple baselines and provide enough ablation studies. Their method performs better than the baselines in terms of satisfying the fairness violation, and the AUC and accuracy values are similar to the baseline models. They also report the frequency of returning NSF.
- The theoretical results are supported by clear assumptions and easy-to-follow proofs.

Weaknesses
- The adversarial inference utilizes the relationship between the DP gap and the covariance between the predicted sensitive attribute and the true sensitive attribute. It is not clear to me how this will generalize for other fairness notions, such as equal opportunity or equalized odds.
- The student-t confidence bound relies on CLT; there is no study on how this affects small per-group sample sizes. The CLT approximation could behave poorly.
- In the experiments section, the FRG and baseline methods use the same number of training samples. The baselines use another dataset for validation, but FRG uses this dataset for both validation and fairness testing. I think that for a fair comparison, the fairness-test dataset should be picked among the training samples.

Other comments:
- The paper is well written, except for a few parts. I think that the section confidence upper bound on fairness violation(5.2.1) is a bit confusing; the pairing part can be explained better. Also, the part where the confidence bound hyperparameter for training, $\alpha$, can be explained better, is it just the previous $\hat{U}=\alpha U$?
- Line 206, mulitclass --> multiclass
- Line 259, hyperparamtr --> hyperparameter
- Line 648, reference to equation 32 doesn't look correct to me.

---

> ### Author Rebuttal · Authors · 2025-07-30
>
> Thank you so much for carefully reviewing our work and appreciating our novelty, the empirical and theoretical results. We address your concern as follows.
>
> ---
>
> ## Q1: How will the relationship between $\Delta_{\text{DP}}$ and $|\text{Cov}(\hat{Y}, S)|$ generalize to equal opportunity or equalized odds?
> Thanks for raising this question. We proved in Theorem 5.2 that $\Delta_{\text{DP}}(\tau,\phi) =|\text{Cov}(\hat{Y}, S)|/\text{Var}(S)$. This ***can be extended to equal opportunity (EOP) and equalized odds (EOD) with simple modification***,  but it requires that the outcome labels, Y, for the downstream task of concern are available.
>
> EOP difference can be defined as $\Delta_{\text{EOP}}(\tau,\phi) \coloneqq |\Pr(\hat{Y} = 1 |S =1,Y=1) -  \Pr(\hat{Y} = 1 | S = 0,Y=1)|$. Thus, this is equivalent to $\Delta_{\text{DP}}(\tau,\phi|Y=1)$. Then, following the same proof, we can conclude that $\Delta_{\text{EOP}}(\tau,\phi) = |\text{Cov}(\hat{Y}, S | Y = 1)|/\text{Var}(S|Y=1)$. That is, the same relationship holds under the additional condition that $Y=1$.
>
> Similarly, EOD difference can be defined as $\Delta_{\text{EOD}}(\tau,\phi) \coloneqq \max(|\Pr(\hat{Y} = 1 |S =1,Y=0) -  \Pr(\hat{Y} = 1 | S = 0,Y=0)|, |\Pr(\hat{Y} = 1 |S =1,Y=1) -  \Pr(\hat{Y} = 1 | S = 0,Y=1)|)$  which is equivalent to $\max(\Delta_{\text{DP}}(\tau,\phi|Y=0), \Delta_{\text{DP}}(\tau,\phi|Y=1))$. It is also equivalent to $\max(|\text{Cov}(\hat{Y}, S | Y = 0)|/\text{Var}(S|Y=0), |\text{Cov}(\hat{Y}, S | Y = 1)|/\text{Var}(S|Y=1))$.
>
> We will make sure this is thoroughly discussed in our revision.
>
> ---
>
> ## Q2: The student-t confidence bound relies on CLT; there is no study on how this affects small per-group sample sizes. The CLT approximation could behave poorly.
> Thanks for raising this question. This is a reasonable concern when the data size is extremely small. While student-t assumes normality, there is a study that shows that it is relatively robust to violations of this assumption, even with small sample sizes, as long as the data are not severely skewed or contain extreme outliers [1]. In our experiment, we show that the guarantee is robust to various data sizes (41k to 195k). However, if this assumption is violated, our framework is *flexible to other choices of statistical tools* such as Bootstrap Confidence Intervals, as mentioned in Appendix F for constructing the confidence bounds.
>
> We will extend the discussions on this assumption in our revision.
>
> ---
>
> ## Q3: The baselines use another dataset for validation, but FRG uses this dataset for both validation and fairness testing. I think that for a fair comparison, the fairness-test dataset should be picked among the training samples.
> Thank you for the suggestion. We agree this is a potential weakness in our original setup. We have now completed this new experiment, where the fairness-test set $D_f$ is created from a 10% split of the training data, while keeping all other settings unchanged. The key finding is that our results are robust to this change. The accuracy and fairness outcomes are nearly identical to our original findings (Figure 2-5). We believe that the main reason for this is that the final unseen test set was used identically and fairly across all methods under the old setup. Since the ultimate measure of performance and fairness generalization is on this test set, our central conclusions about FRG's effectiveness hold. We will update the experiment figures accordingly in our revision.
>
>
> ---
>
> ## Q4: The pairing part in Sec. 5.2.1. can be explained better. The part where the confidence bound hyperparameter for training, α, can be explained better, is it just the previous $\hat{U}=\alpha U$?
> Thanks for the suggestions for improving the clarify of our approach.
> Regarding the pairing approach, we essentially want to retrieve $m$ pairs of unbiased estimates for each of $\Pr(\hat{Y} = 1|S = 0)$ and $\Pr(\hat{Y} = 1|S = 1)$. They can form $m$ unbiased point estimates of $\Pr(\hat{Y} = 1|S = 0) - \Pr(\hat{Y} = 1|S = 1)$. These point estimates will be used to construct confidence intervals on $\Pr(\hat{Y} = 1|S = 0) - \Pr(\hat{Y} = 1|S = 1)$ by applying statistical tools such as Student’s t-test. We will break down these steps further to make it more digestible.
>
> You are correct that $\hat{U}=\alpha U$. The hyperparameter $\alpha$ is used to inflate the high-confidence upper bound $U$ such that the candidate selection is less likely to overfit to $U$, which may result in more NSF. We will clarify this in our manuscript.
>
> ---
>
> We thank you again for your help in improving our manuscript. We will incorporate your other comments in our revision.

---

> > ### Comment · Reviewer_XhGo · 2025-08-04
> >
> > Thanks for the response and the clarifications. I am happy to see that the results remain unchanged when the fairness testing dataset is split from the training dataset. I think that the proposed explanation for the pairing will clear that part. I appreciate the explanation regarding how the covariance relationship will generalize to other fairness notions. However, I am also curious whether this covariance relationship will generalize to sensitive attributes that aren't just binary; a more general result could strengthen the paper.
> >
> > I will maintain my original score for this paper for now.

---

> > > ### Author Response · Authors · 2025-08-05
> > > **Response to the question about generalizing the covariance relationship for non-binary sensitive attributes**
> > >
> > > Thank you for reviewing our rebuttal and providing positive feedback. In particular, we thank you for raising the following insightful question. We address it below.
> > >
> > > ## How can the relationship between $\Delta_{\text{DP}}$ and $|\text{Cov}(\hat{Y}, S)|$ be extended to non-binary sensitive attributes?
> > >
> > > The standard definition of covariance is not applicable to non-binary categorical random variables like the sensitive attributes. The reason is that the covariance takes the numerical values of the random variable into account but the numerical values of the sensitive attribute have no actual meaning.
> > >
> > > However, we can define auxiliary random variables for $S$ for each pair of sensitive categories $i,j \\in \\mathcal{S}$, to represent $S$ as a set of binary indicator variables, such that covariance can be applied. This approach enables the application of FRG to the setting of non-binary sensitive attributes $S$.
> > >
> > > Suppose $S\\in \\mathcal{S}$ where $|\mathcal{S}| > 2$. Create one indicator variable $S^\prime_{i,j}$ for each pair of $i,j \\in \\mathcal{S}$ where $i\\ne j$ such that $S^\prime_{i,j} = 0$ if $S = i$ and $S^\prime_{i,j} = 1$ if $S = j$.
> > >
> > > We denote $p_{a,b} \coloneqq \Pr[\hat{Y}=a, S = b]$ where $a \in \\{0, 1\\}$ and $b\in \mathcal{S}$. We will first prove the lemma below before stating the main theorem.
> > >
> > > **Lemma 1.** $\text{Cov}(\hat{Y},S^\prime_{i,j}|S\in\\{i,j\\}) = \frac{p_{1,j}p_{0,i} - p_{1,i}p_{0,j}}{(\Pr(S = i) + \Pr(S = j))^2}$.
> > >
> > >
> > > **Proof.** Following the same proof as Lemma B.1 in Appendix B, we have
> > >
> > > $\text{Cov}(\hat{Y},S^\prime_{i,j}|S\in\\{i,j\\})$
> > >
> > > &emsp;$=
> > > \Pr[\hat{Y} = 1, S^\prime_{i,j} = 1 | S\in\\{i,j\\}]\Pr[\hat{Y} = 0, S^\prime_{i,j} = 0 | S\in\\{i,j\\}]$
> > >
> > > &emsp;&emsp;$- \Pr[\hat{Y} = 1, S^\prime_{i,j} = 0 | S\in\\{i,j\\}]\Pr[\hat{Y} = 0, S^\prime_{i,j} = 1 | S\in\\{i,j\\}]$
> > >
> > > &emsp;$=\Pr[\hat{Y} = 1, S = j | S\in\\{i,j\\}]\Pr[\hat{Y} = 0, S = i | S\in\\{i,j\\}]$
> > >
> > > &emsp;&emsp;$- \Pr[\hat{Y} = 1, S = i | S\in\\{i,j\\}]\Pr[\hat{Y} = 0, S = j | S\in\\{i,j\\}].$
> > >
> > > Since $\Pr[S\in\\{i,j\\}] = \Pr(S=i) + \Pr(S=j)$, $\text{Cov}(\hat{Y},S^\prime_{i,j}|S\in\\{i,j\\}) = \frac{p_{1,j}p_{0,i} - p_{1,i}p_{0,j}}{(\Pr(S = i) + \Pr(S = j))^2}$. This completes the proof.
> > >
> > > Demographic disparity can be defined separately for each pair of sensitive categories, $i,j \in \mathcal{S}$, as $\Delta_{\text{DP}}^{i,j} = \Big |\Pr(\hat{Y} = 1 | S = i) -  \Pr(\hat{Y} = 1 | S = j)\Big |$. Then, we can limit $\Delta_{\text{DP}} = \max_{i,j} \Delta_{\text{DP}}^{i,j}$, to ensure demographic parity with respect to any pair of sensitive categories (Bird et al. [2]).
> > > Next, we provide the relationship between $\Delta_{\text{DP}}$ and $\text{Cov}(\hat{Y},S^\prime_{i,j}|S\in\\{i,j\\})$.
> > >
> > > **Theorem 1.2.**
> > > $\Delta_{\text{DP}}(\tau, \phi) = \max_{i,j} \Big(2 + \frac{\Pr(S = i)}{\Pr(S = j)} + \frac{\Pr(S = j)}{\Pr(S = i)}\Big) \Big|\text{Cov}(\hat{Y}, S^\prime_{i,j}| S \in\\{i,j\\}) \Big |,$ where $i,j\in \mathcal{S}$ and $i\ne j$.
> > >
> > > **Proof.**
> > >
> > > $\Delta_{\text{DP}}(\tau, \phi) $
> > >
> > > &emsp;$= \max_{i,j} \Big|\Pr(\hat{Y} = 1 | S = i) -  \Pr(\hat{Y} = 1 | S = j)\Big| $
> > >
> > > &emsp;$= \max_{i,j} \Big |\frac{\Pr(\hat{Y} = 1, S = i)}{\Pr(S = i)} - \frac{\Pr(\hat{Y} = 1, S = j)}{\Pr(S = j)}\Big |$
> > >
> > > &emsp;$= \max_{i,j} \Big |\frac{\Pr(\hat{Y} = 1, S = i)\Pr(S = j) - \Pr(\hat{Y} = 1, S = j)\Pr(S = i)}{\Pr(S = i)\Pr(S = j)}\Big |$
> > >
> > > &emsp;$= \max_{i,j} \frac{1}{\Pr(S = i)\Pr(S = j)}\Big | p_{1,i}p_{0,j} + p_{1,i}p_{1,j} - p_{1,j}p_{0,i} - p_{1,j}p_{1,i} \Big | $
> > >
> > > &emsp;$= \max_{i,j} \frac{1}{\Pr(S = i)\Pr(S = j)}\Big | p_{1,i}p_{0,j} - p_{1,j}p_{0,i} \Big | $
> > >
> > > &emsp;$= \max_{i,j} \frac{(\Pr(S = i) + \Pr(S = j))^2}{\Pr(S = i)\Pr(S = j)}\Big | \frac{p_{1,i}p_{0,j} - p_{1,j}p_{0,i}}{(\Pr(S = i) + \Pr(S = j))^2} \Big | $
> > >
> > > &emsp;$= \max_{i,j} \Big(2 + \frac{\Pr(S = i)}{\Pr(S = j)} + \frac{\Pr(S = j)}{\Pr(S = i)}\Big) \Big|\text{Cov}(\hat{Y}, S^\prime_{i,j}| S \in\\{i,j\\}) \Big |$
> > >
> > > This completes the proof. Using this relationship, the optimal adversary can be approximated for non-binary sensitive features.
> > >
> > > For instance, suppose that $\Pr(S=i) = \frac{1}{|\mathcal{S}|}$ for all $i$. Then, $\Delta_{\text{DP}}$ is minimized when the predicted label $\hat{Y}$ does not provide any information differentiating any pair of $i$ and $j\in \mathcal{S}$. On the other hand, $\Delta_{\text{DP}}$ is maximized if there exists a pair of $i$ and $j$ such that $\hat{Y}$ is maximally correlated with $S$ conditioning on $S\in \\{i,j\\}$.
> > >
> > > We hope that you find this analysis clear. Thank you again for your review!
> > >
> > > #### [2] Bird et al. Fairlearn: A toolkit for assessing and improving fairness in ai. Microsoft, 2020.

---

> > > > ### Comment · Reviewer_XhGo · 2025-08-05
> > > >
> > > > Thanks for the detailed explanation regarding how non-binary sensitive attributes can be handled.

---

### Official Review · Reviewer_yn4p · 2025-07-03

**Clarity:** 3
**Significance:** 2
**Originality:** 3
**Rating:** 5
**Confidence:** 3

**Summary:**

This paper introduces Fair Representation learning with high confidence Guarantees (FRG), which is an algorithm for learning models that are guaranteed to be fair in terms of demographic parity when applied to downstream tasks with high probability. The high-level idea applied by FRG is to first train a candidate model (e.g. a variational autoencoder modified with a fairness constraint) using part of the data and then train an adversary that aims to predict protected attributes using the representations from the candidate model. The adversary is used to check whether the candidate model provides the guarantees specified by the user. More specifically, the adversary is used to estimate the fairness bounds based on small data samples (data pairs) using a statistical test (e.g., student-t). Under some assumptions on the data samples, the number of samples, and the accuracy of the adversary, the paper shows that the theoretical guarantee on fairness is achieved. In the experiments, FRG is compared against six baselines in terms of downstream fairness and accuracy using three datasets. The results show that FRG achieves better accuracy than alternatives that are able to guarantee downstream fairness with high probability.

**Questions:**

Q1. Are there non-trivial fairness vs. quality tradeoffs with the guarantees provided in the paper? Is it possible to provide a more comprehensive view of such tradeoffs to demonstrate the practical use of the proposed solution?

Q2. How realistic are the assumptions considered in the paper and how robust is the method to small violations of such assumptions that might occur in real datasets.

Q3. How is Theorem 5.2 useful in the paper?

**Ethical Concerns:**

["NO or VERY MINOR ethics concerns only"]

**Final Justification:**

I believe this paper makes an interesting contribution towards fair machine learning by incorporating confidence guarantees into fairness assessments. My original review was positive (accept) and the authors have clarified my questions during the rebuttal. My suggestions for improving the paper were the following:

1) Adding some discussion about what values of confidence lead to practical accuracy values on real datasets;

2) Providing some intuition on why it is easier to maximize the covariance instead minimizing demographic parity (Theorem 5.2).

**Limitations:**

The limitations are discussed in the paper.

**Paper Formatting Concerns:**

NA.

**Quality:**

4

**Strengths And Weaknesses:**

**Strengths:**
- The paper addresses a relevant problem.
- The paper is well-written and its motivation is clear.
- The experimental results support the main conclusions in the paper.

**Weaknesses:**
- The accuracy of the proposed approach seems too low when fairness bound is tight
- There is little discussion about the assumptions of the proposed model
- It is not clear how theorem 5.2 is used in the paper (compared to DP)

**Detailed comments:**

This is an interesting paper addressing a relevant problem but with some clarifications needed regarding its assumptions and experiments. I will detail my main concerns here.

*Accuracy:* The accuracy results for FRG reported in Figure 2 seem very low when epsilon is small. This might be a limitation of the dataset, the model, or it could be justified theoretically. It would be helpful to understand the source of the error to some extent. In particular, one can assume that there are no practically useful models that can provide the type of guarantees considered in this paper.

*Assumptions:* The main assumptions of the proposed approach are listed in the limitations but some discussion is needed regarding how realistic these assumptions are and whether they are expected to hold for the datasets considered. In particular, the assumption of an optimal adversary seems unrealistic, so a stronger result would bound the bounds in terms of how close the adversary is to optimal or in terms of the adversary model and the data used to train it. A synthetic dataset could be useful to clarify this.

*Theorem 5.2:* It was not clear to me why Theorem 5.2 is useful in this paper. Does it provide any benefit compared with the original definition of Demographic Parity?

---

> ### Author Rebuttal · Authors · 2025-07-30
>
> Thank you so much for carefully reviewing and appreciating our work and raising insightful questions. We address your concern as follows.
>
> ---
>
> ## Q1: The accuracy of the proposed approach seems too low when the fairness bound is tight (Figure 2).
> We think your comment is insightful. On the Adults dataset, when the fairness constraint is strict, e.g., $\epsilon <= 0.08$, the AUCs of our method are around 60-63% which are *better than all baselines that satisfy the constraints* (ICVAE and FARE). Referring to Appendix Figure 6, we can see that our method achieves the best fairness and accuracy tradeoff. So we agree with you that one can assume there are no practically useful models that can provide such a strict guarantee. In this case, we think it is due to a small $\epsilon$.
>
> ---
>
> ## Q2: There is little discussion about the assumptions of the proposed model. How realistic these assumptions are and whether they are expected to hold for the datasets considered?
> Thanks for the comments. Our main assumptions are provided in Section 7 which include (1) data samples are i.i.d., (2) Student's t-test assumes CLT holds, (3) access to an optimal adversary is available.
>
> Regarding (1), the guarantee could be sensitive to some distributional shifts. If the test dataset has a different distribution of (X,S), the guarantee that the representation model is fair with high probability may no longer hold. However, the guarantee is robust to label shifts because the label is not necessary when learning the representation model and constructing the high-confidence upper bounds. That is, if the test dataset has a different distribution of label Y or distribution of Y conditioned on (X,S), the guarantee will still hold.
>
> Regarding (2), we think this assumption is not very sensitive for most datasets. Our guarantee seems robust for datasets of various sizes (from 41k to 195k). While the t-test assumes normality, it is relatively robust to violations of this assumption, even with small sample sizes, as long as the data are not severely skewed or contain extreme outliers [1].
>
> Regarding (3), we think the choice of the adversary **will** affect the robustness of the guarantee. One should choose an adversary that achieves high performance on the adversarial task. If a weak model is chosen, it will not approximate the optimal adversary well. In general, we think the best choice of the adversary would vary case by case and be informed by the prediction performance on the adversarial task. While these notes are purely empirical, how to provide a theoretical bound on optimal adversary is still an active research problem and could be studied by future works (this is discussed in the response to Question 1 by Reviewer xgom).
>
> We will provide an extended discussion of each assumption in our revision.
>
> #### [1] de Winter, J., Using the Student's t-test with extremely small sample sizes, Practical Assessment, Research, and Evaluation 18(1): 10. 2013.
>
> ---
>
> ## Q3: It is not clear how Theorem 5.2 is used in the paper. What is the benefit compared to $\Delta_{\text{DP}}$?
> Thanks for pointing out that this was not clear. Theorem 5.2 is not used to prove that FRG provides the high-confidence fairness guarantees. However, it inspires our choice of the adversarial objective. We define the optimal adversary to be one that maximizes $\Delta_{\text{DP}}$. In Theorem 5.2, we show that maximizing $\Delta_{\text{DP}}$ is equivalent to maximizing $|$Cov$(\hat{Y} , S)|$. Thus, this gives justification that optimizing the adversary with the objective of maximizing $|$Cov$(\hat{Y} , S)|$ can approximate the optimal adversary. Empirically, we show that our design of the adversarial prediction is effective in providing the desired guarantees. We will be sure to clarify this in our revision.
>
>
> ---
>
> ## Q4: Are there non-trivial fairness vs. quality tradeoffs with the guarantees provided in the paper?
> Thanks for raising this question. The fairness vs utility tradeoff for each dataset is provided in Appendix Figure 6. Our method achieves competitive tradeoffs compared to the baselines. Specifically, on the Adults dataset, FRG achieves the pareto frontier of this tradeoff.
>
> ---
>
> Thank you again for reviewing the work, and we will be happy to discuss any further questions you may have.

---

> > ### Comment · Reviewer_yn4p · 2025-08-04
> > **Response to the rebuttal**
> >
> > Thanks for the response. I suggest adding some discussion about what values of confidence lead to practical accuracy values on real datasets. Regarding Q3, some intuition on why it is easier to maximize the covariance could also be helpful. My review is already positive, so I will maintain my original recommendation for this paper.

---

> > > ### Author Response · Authors · 2025-08-04
> > > **Thank you!**
> > >
> > > Thank you for responding to our rebuttal. We appreciate your suggestions, which will certainly help us improve the manuscript. Thank you for recommending our work.

---

### Official Review · Reviewer_xgom · 2025-07-03

**Clarity:** 3
**Significance:** 3
**Originality:** 2
**Rating:** 4
**Confidence:** 4

**Summary:**

This paper formally introduces the task of learning representations that achieve high-confidence fairness. Then it presents a novel framework FRG that learns fair representations for all downstream tasks based on demographic parity.  It guarantees with high confidence that the output representation models are fair according to user-specified thresholds of unfairness and confidence levels. Experiments on three real-world datasets demonstrate that FRG consistently bounds unfairness across a range of downstream models and tasks.

**Questions:**

Besides the questions in the Weaknesses, I also have the following questions:
- Does the proposed method apply to other supervised fairness metrics, such as equal opportunity and equalized odds?
- What are the computational tradeoffs? Can you compare training time with otherbaselines?
- The details of the model architecture and parameter settings are missing in the paper.
- In line 189, you mentioned "In practice (section 6), we find it sufficient to train an approximately optimal adversary to predict S based on Z". Can you elaborate more? Does the choice of adversary design affect this statement?

**Ethical Concerns:**

["NO or VERY MINOR ethics concerns only"]

**Final Justification:**

My concerns on the adversary choice, questions on the training cost, and generalization to other fairness metrics are properly addressed.

**Limitations:**

Yes

**Paper Formatting Concerns:**

I have no concern regarding the format.

**Quality:**

3

**Strengths And Weaknesses:**

Strengths:

- This paper identifies a practical gap, a lack of high-confidence guarantees in current fair representation learning approaches. And it extends fairness guarantees to all downstream tasks and models. The studied problem is interesting and significant.
- Experiments on the three datasets and a range of tasks demonstrate the effectiveness of the proposed framework in both fairness and useful representation learning.
- The paper is clearly written and easy to follow.

Weaknesses:
- The theoretical analysis in section 5.2.3 assumes access to an optimal adversary. Although FRG approximates the adversary, it might not always be realistic in practice. Is there theoretical support for the guarantee of obtaining the optimal adversary, such as requirements for model complexity and training data size?
- Repeated training of adversaries and statistical testing introduces a computational burden. While not prohibitive, this could limit real-world scalability. The author should also present a comparison of the computation cost/time.
- The presentation of the experimental results is poor. In Figure 3, I can hardly see the blue line, which is the performance of the proposed method.
- The choice of datasets is limited, and some of them are outdated. For the adult dataset, you should better use this updated one: "Retiring Adult: New Datasets for Fair Machine Learning"

---

> ### Author Rebuttal · Authors · 2025-07-30
>
> We thank Reviewer xgom for carefully reviewing our work and appreciating the significance of the studied problem and the delivery of the writing and experiment. We provide detailed discussions below to address your concerns.
>
> ---
>
> ## Q1: The assumption of access to an optimal adversary might not be realistic. Is there theoretical support for the guarantee of obtaining the optimal adversary?
> Our theoretical analysis does assume access to an optimal adversary, as pointed out throughout the paper. We address this concern as follows.
>
> First, we *empirically validate that our adversary is a sufficiently strong proxy for the optimal one*. The adversary is approximated by training an MLP to maximize $|$Cov$(\hat{Y}, S)|$ (Sec. 5.1). This architecture, despite its simplicity, achieves very high AUC, > 98% AUC for each dataset, on the adversarial task when representations are learned without fairness considerations. Our experimental results suggest that the adversary achieved with gradient optimization is sufficient to obtain a valid high-confidence upper bound on the unfairness of arbitrary downstream tasks.
>
> Second, a guarantee for an optimal adversary *can be provided by restricting the model and the representation complexities*. This approach was taken by FARE[2], which replaces the representation model with the fair decision tree to restrict the representation to be drawn from a discrete distribution with a finite support. While this approach can guarantee access to the "optimal adversary" defined in [2] and provide a theoretically guaranteed upper-bound on $\Delta_{\text{DP}}$, the limitations of FARE will be carried over. In experiment (Sec. 6.2) , we found that FARE can only produce limited variations of representations, so it cannot control the tradeoff between performance and fairness with enough granularity. A lot of times, it is impossible to find a representation model that satisfies the fairness constraints when $\epsilon$ is small, while it is still possible for FRG (Figures 2 and 3). Essentially, this approach violates the controllability, which limits the practicality.
>
> Third, we note that the question of how to provide theoretical guarantees for obtaining or bounding the performance of optimal adversary is *a common challenge faced by the field*. For example, Feng et al. [7] derived a bound to the accuracy of the optimal Lipschitz-continuous adversary, but the bound is non-tractable in practice. Similar assumptions of access to optimal adversaries have been used extensively by previous works (e.g., [1-6]). Thus, we believe a solution to this challenge merits its own study.
>
> Finally, we also *developed an approach that does not require optimal adversary*, using a Mutual Information (MI)-based upper-bound on $\Delta_{\text{DP}}$ (Appendix K). This approach gives a theoretically guaranteed upper-bound on $\Delta_{\text{DP}}$ across arbitrary downstream models and tasks that does not assume optimal adversary. However, experiment shows that this upper-bound can be significantly loose (Figure 12) and thus enforcing fairness constraint on this upper-bound is non-practical (Figure 13).
>
> Thank you for raising this important question. We will incorporate a summary of this discussion in the limitations section in our revision.
>
> ---
>
> ## Q2: Does the choice of adversary design affect the guarantee (the claim in line 189)?
> Thank you so much for pointing out that this was not clear. We think the choice of the adversary **will** affect the robustness of the guarantee. One should choose an adversary that achieves high performance on the adversarial task. If a weak model is chosen, it will not approximate the optimal adversary well. In our implementation, we use an MLP with one hidden layer and tunable hidden sizes as the adversary. We think this architecture is strong enough for the datasets we have tested. We have also verified that this architecture will lead to high prediction performance (> 98% AUC across our three datasets) of the adversarial task if the representations are learned without fairness consideration. We think the optimal adversary is well approximated such that even on the adversarial downstream tasks, we can still guarantee fairness while the baselines cannot. In general, we think the best choice of the adversary would vary case by case and be informed by the prediction performance on the adversarial task. While these notes are purely empirical, how to provide a theoretical bound on optimal adversary is still an active research problem as discussed in Question 1. We will make sure to clarify the choice of the adversary design in our revision.
>
> ---
>
> ## Q3: Repeated training of adversaries and statistical testing introduces a computational burden. What is the computation cost compared to baselines?
> While the training of the adversary and the fairness test introduce some computational overhead, they are *very small* (around 5% of the overall runtime) compared to the training time of the representation model (95% of the runtime). Here is an example for the Income dataset. We train the representation models with 500 epochs across all baselines. The overall runtime of FRG is 330s, which includes training the adversary, representation model, and the fairness test. Training the adversarial model takes 16s (4.8%), the fairness test takes only 0.06s (0.02%), while training the representation takes about 95% of the runtime. Here is a comparison with the baselines: CFAIR takes 303s, ICVAE takes 1021s, LMIFR takes 336s, LAFTR takes 326s, FARE takes 41s (only one call to fit the decision tree). Thus, *we think the computation cost is not a concern*. In the revised manuscript, we will include a short paragraph clarifying this.
>
> ---
>
> ## Q4: The presentation of the experimental results is poor. In Figure 3, the blue line is hard to see.
> Thanks for raising this concern. We *have increased the marker sizes and used new marker shapes for FRG* and they appear much more visible now. We will make sure the final version shows the results as clearly as possible. To address your concern, the blue line achieves very similar results to the orange line, which is also our proposed approach but with supervised loss, thus they overlap. They always satisfy the fairness constraints.
>
> ---
>
> ## Q5: The choice of datasets is limited, and some of them are outdated. For the adult dataset, you should better use this updated one: "Retiring Adult: New Datasets for Fair Machine Learning"
> Thanks for the suggestion. To clarify, ***we have included the new Retiring Adult dataset you mentioned***. We call the new dataset the "Income" dataset (citation [17] in the manuscript) to differentiate from the traditionally used "Adults" dataset. In total, we have included three distinct datasets, each with 2-3 downstream evaluations, which matches most previous works. We will clarify this naming convention in the revision to avoid any confusion.
>
> ---
>
> ## Q6: Does the proposed method apply to other supervised fairness metrics, such as equal opportunity and equalized odds?
> This is an insightful question. Indeed, the proposed method ***can be applied to Equal Opportunity (EOP) and equalized odds (EOD) with simple modification***, but it requires that the outcome labels, Y, for the downstream task of concern are available to compute EOP and/or EOD. We will use EOP as an example. $\Delta_{\text{EOP}}(\tau,\phi) \coloneqq |\Pr(\hat{Y} = 1 |S =1,Y=1) -  \Pr(\hat{Y} = 1 | S = 0,Y=1)|$. This is equivalent to $\Delta_{\text{DP}}(\tau,\phi|Y=1)$. The rest follows, except we only consider the subset of the data where $Y=1$ when constructing the high-confidence upper-bound. Similarly, $\Delta_{\text{EOD}}(\tau,\phi) = \max(\Delta_{\text{DP}}(\tau,\phi|Y=0), \Delta_{\text{DP}}(\tau,\phi|Y=1))$ which one can also construct the high-confidence upper-bound using our framework by splitting $\delta$ in half for each of $\Delta_{\text{DP}}(\tau,\phi|Y=0)$ and $\Delta_{\text{DP}}(\tau,\phi|Y=1).$
>
> In the revised manuscript, we will clarify how our approach can be used to constrain EOP and EOD. However, we note that the high-confidence upper bound requires outcome labels, Y, from a specific downstream task, so in this context the guarantee can be applied only to that task, which *might not be generalized to arbitrary downstream tasks*, as is the case for the DP (demographic parity). This is a common limitation to existing approaches as well (e.g.,[2-3]).
>
> ---
>
> ## Q7: The details of the model architecture and parameter settings are missing in the paper.
> Sorry that the details of the parameter settings are not included in the paper. We use different hyperparameters for different $\epsilon$'s and datasets. We included all of the hyperparameter choices for each of the settings in the config JSON files in the source code provided (as mentioned in Appendix H “Hyperparameter Tuning” referenced in the main text, line 333). The model architecture of our method is discussed in Sec. 5, and the downstream model architecture is discussed in Appendix H. To improve readability, we will also add a summary table of the key hyperparameters choices to the appendix in our revision.
>
> ---
>
> Thank you again for reviewing the work, and we will be happy to discuss any further questions you may have.
>
> #### [1] Gupta et al. Controllable guarantees for fair outcomes via contrastive information estimation. AAAI 21.
>
> #### [2] Jovanović et al. Fare: Provably fair representation learning with practical certificates. ICML 23.
>
> #### [3] Song et al. Learning controllable fair representations. AISTATS 19.
> #### [4] Madras et al. Learning adversarially fair and transferable representation. ICML 18.
>
> #### [5] Kairouz et al. Generating fair universal representations using adversarial models. IEEE TIFS 22.
>
> #### [6] Zhao et al. Conditional learning of fair representations. ICLR 20.
>
> #### [7] Feng et al. Learning fair representations via an adversarial framework. CoRR 19.

---

> > ### Author Response · Authors · 2025-08-05
> > **Discussion regarding the rebuttal**
> >
> > Dear Reviewer xgom,
> >
> > We are writing to follow up on our rebuttal. As we are approaching the end of the discussion period, we would like to check if our response has sufficiently addressed the concerns you raised. We would be grateful for any feedback and are happy to answer any further questions.
> >
> > Thank you so much for your help in improving our work.

---

> > ### Comment · Reviewer_xgom · 2025-08-07
> >
> > Thank you for the response. Most of my questions and concerns are addressed properly. Please include some discussions and clarifications in the revision. I have updated my score to reflect my assessment of the work.

---

> > > ### Author Response · Authors · 2025-08-08
> > > **Thank you!**
> > >
> > > Thank you for taking the time to read and respond to our rebuttal and for your positive feedback. We will be sure to incorporate the discussions and clarifications above in the final version.

---

### Decision · Program_Chairs · 2025-09-17

**Decision:**

Accept (poster)

**Comment:**

This submission proposes FRG, a framework for fair representation learning with controllable high-confidence guarantees via adversarial inference. The paper received consistently positive ratings from all reviewers who recognized its novel approach to providing statistical guarantees for fairness across arbitrary downstream tasks.

Reviewers initially raised concerns about the optimal adversary assumption, computational overhead, and experimental setup. The authors provided thorough responses including empirical validation of adversary quality, demonstration of minimal computational overhead, and additional experiments with modified data splits showing robust results. The theoretical extensions to other fairness metrics and non-binary attributes further strengthen the work's applicability.

The work addresses an important gap in fair representation learning by providing statistical guarantees rather than empirical estimates, with practical controllability through user-specified thresholds. The theoretical framework is sound, the empirical validation is comprehensive, and the author responses demonstrate thorough engagement with reviewer feedback.

I recommend **ACCEPT**.